Review

behaviour, ecology, evolution

wealth inequality, niche construction, social evolution, social mobility, intergenerational wealth transmission, status-seeking behaviour

**Author for correspondence:**
Eli D. Strauss
e-mail: estrauss@ab.mpg.de

# The ecology of wealth inequality in animal societies

Eli D. Strauss[1,2,3,4] and Daizaburo Shizuka[3]

[1]Department of Collective Behaviour, Max Planck Institute of Animal Behaviour, Konstanz, Germany
[2]Centre for the Advanced Study of Collective Behaviour, University of Konstanz, Konstanz, Germany
[3]School of Biological Sciences, University of Nebraska-Lincoln, Lincoln, NE, USA
[4]BEACON Center for the Study of Evolution in Action, Michigan State University, Lansing, MI, USA

 EDS, 0000-0003-3413-1642; DS, 0000-0002-0478-6309

Individuals vary in their access to resources, social connections and phenotypic traits, and a central goal of evolutionary biology is to understand how this variation arises and influences fitness. Parallel research on humans has focused on the causes and consequences of variation in material possessions, opportunity and health. Central to both fields of study is that unequal distribution of wealth is an important component of social structure that drives variation in relevant outcomes. Here, we advance a research framework and agenda for studying wealth inequality within an ecological and evolutionary context. This ecology of inequality approach presents the opportunity to reintegrate key evolutionary concepts as different dimensions of the link between wealth and fitness by (i) developing measures of wealth and inequality as taxonomically broad features of societies, (ii) considering how feedback loops link inequality to individual and societal outcomes, (iii) exploring the ecological and evolutionary underpinnings of what makes some societies more unequal than others, and (iv) studying the long-term dynamics of inequality as a central component of social evolution. We hope that this framework will facilitate a cohesive understanding of inequality as a widespread biological phenomenon and clarify the role of social systems as central to evolutionary biology.

## 1. Introduction

Inequality is a general feature of human and non-human animal societies. Most societies exhibit disparities in individual access to resources, physical condition and social relationships. These disparities can be conceptualized as dimensions of wealth inequality, which translate into differences in outcomes such as health, longevity and reproductive success, and ultimately influence variation in fitness. Wealth inequality in different dimensions may be driven by similar underlying processes and have shared effects on outcomes. Social systems may also differ in which dimension of wealth most directly influences individual outcomes. An overarching study of the causes and consequences of wealth inequality facilitates comparisons of the mechanisms underlying variation in outcomes in various societies. Such a perspective can interrogate the myriad potential factors that generate and maintain wealth inequality, scrutinize the consequences of wealth inequality in terms of individual health and reproductive outcomes, or investigate how inequality changes across time within a society.

Researchers in both human- and animal-oriented fields are motivated to understand how wealth inequality arises, is sustained and acts as a mechanism underlying disparities in outcomes, but the general emphasis differs across fields. In the study of modern human societies, research often focuses on how wealth inequality influences health and well-being, with the aim of informing policies that reduce disparities and promote the well-being of as many people as possible. Research in evolutionary anthropology and related fields examines the role of inequality in human evolution, including the evolutionary origins of human

societies and the effects of inequality on fitness in humans [1–7]. In studies of animal societies, the focus often takes an explicitly evolutionary biology perspective, focusing on wealth inequality as a mechanism that generates variation in fitness.

Wealth, inequality and their influences on fitness variation have been considered in different contexts within the fields of evolution and ecology. For instance, a century of work has explored how networks of dominance relationships arise from interactions among group-mates and influence social structure and fitness-related outcomes [8]. Sexual selection theory addresses the causes and consequences of inequality in mating success [9], and studies of reproductive skew examine behavioural constraints on inequality in reproduction [10,11]. Research into collective decision-making explores the causes and consequences of inequality in behavioural decisions [12–14]. Woven into these subfields are theories of kin selection and multilevel selection, which seek to identify how individual wealth influences the indirect fitness of other individuals, and how inequalities within and between groups influence evolution. Thus, much work on social evolution has concerned itself with the causes and consequences of wealth inequality, albeit without explicitly referring to the parallel concepts of wealth and inequality that human-oriented fields have more thoroughly explored. Notable exceptions are work on privatization and property by Strassman & Queller [15] and intergenerational wealth transfer by Smith *et al.* [16]. In this paper, we expand on this prior work to provide a more overarching review of the concepts of wealth and inequality in animal societies, and explore how wealth inequality can be a source of social selection [17–19].

Here we present a research agenda for studying wealth inequality within an ecological and evolutionary context. We synthesize concepts, questions and empirical insights from research in animals and humans to investigate the ecological and evolutionary implications of inequality. We show that this 'ecology of wealth inequality' approach presents the opportunity to clarify the role of social systems as central to evolutionary biology, and to reintegrate key evolutionary concepts that have often been perceived as alternatives (e.g. trait evolution, niche construction, extended phenotypes) as different dimensions of the wealth–fitness relationship. We identify four key opportunities in the ecological study of inequality: (i) developing measures of wealth and inequality as taxonomically broad features of societies, (ii) considering how feedback loops link inequality to individual and societal outcomes, (iii) exploring the ecological and evolutionary underpinnings of what makes some societies more unequal than others, and (iv) studying the long-term dynamics of inequality as a central component of social evolution. In each section, we review existing work and highlight areas requiring additional empirical and theoretical attention. We aim to motivate a cohesive interdisciplinary approach to understanding inequality as a widespread and diverse biological phenomenon.

## 2. What are wealth and inequality in animal societies?

Non-humans do not have bank accounts, so how can they be wealthy? Economists and evolutionary anthropologists have long known that wealth can take many forms [20,21]. Wealth manifests in many *currencies*, or quantities of attributes or possessions that impact an individual's access to

'valued goods and services' [22]. Although the currencies of wealth are numerous, they can be pooled into three superseding categories (here '*aspects*'; figure 1, top left) [4,22,23]. *Material wealth* denotes extrasomatic currencies such as money, land or livestock. *Relational wealth* consists of social connections, often measured as ties in a network of relevant social interactions or relationships such as food sharing, prestige or cooperative hunting. Finally, *embodied wealth* refers to attributes of individuals, such as size, strength or knowledge.

This framework reveals how animal societies are also structured by multiple dimensions of wealth. These same three aspects—material, relational and embodied wealth—are key elements of animal societies and map clearly onto established concepts in ecology and evolution, such as constructed/defended niches, social niches and phenotypic traits. Material wealth currencies include defendable resources such as food items, nest sites and territories, as well as 'constructed' resources such as food caches, shelters and nest decorations [15,16]. For instance, material wealth is prominent in acorn woodpeckers (*Melanerpes formicivorus*), which invest heavily both in granary construction (the work of generations of woodpeckers) and in the collection and storage of acorns within the granary [24]. Material wealth may also take the form of empty snail shells occupied by hermit crabs (*Pagurus longicarpus*)—resources that are unequally distributed in quality and directly affect fitness outcomes [25]. Relational wealth describes an individual's social niche [26], encompassing social relationships and interactions such as grooming, huddling or dominance. Considerable evidence points to the impact that relational wealth has in human and non-human animal societies [6,27,28]. For example, social alliances influence rank and fitness in spotted hyenas (*Crocuta crocuta*) [29]. Embodied wealth is made up of phenotypic currencies such as body size, fat reserves, sperm quality, ornament size, display quality or information. Classic examples of embodied wealth are condition-dependent signals, such as the male house-finch's (*Carpodacus mexicanus*) bright red plumage [30]. These different aspects of wealth operate concurrently, and biological market theory provides a framework for understanding exchanges in a wealth of different currencies [31].

Wealth inequality describes the spread and skewness of distributions of wealth (figure 1, centre circle) in these different dimensions (box 1). The scale at which inequality is assessed can be tuned flexibly according to the question and the study species. For instance, one can measure inequality among individuals in a society or social group, or among individuals in a population consisting of multiple social groups. When wealth operates at the group level (e.g. group territories, shared food caches), wealth inequality among groups can be assessed at the population level.

There is broad consensus in evolutionary theory that material and relational wealth (i.e. constructed and social niches) can influence fitness, drive adaptation and contribute to evolutionary change [44]. Existing biological concepts also describe the transmission of wealth across generations via mechanisms of genetic and epigenetic inheritance, ecological inheritance [45] and social inheritance [46]. Intergenerational transmission of wealth may affect 'privilege' as a source of inequality in animal societies [16]. Exploring evolutionary themes such as niche construction and social inheritance from the lens of wealth inequality could provide clarity to debates on how to integrate these dynamics in evolutionary theory [47,48]. Specifically, we argue that the patterns of distribution

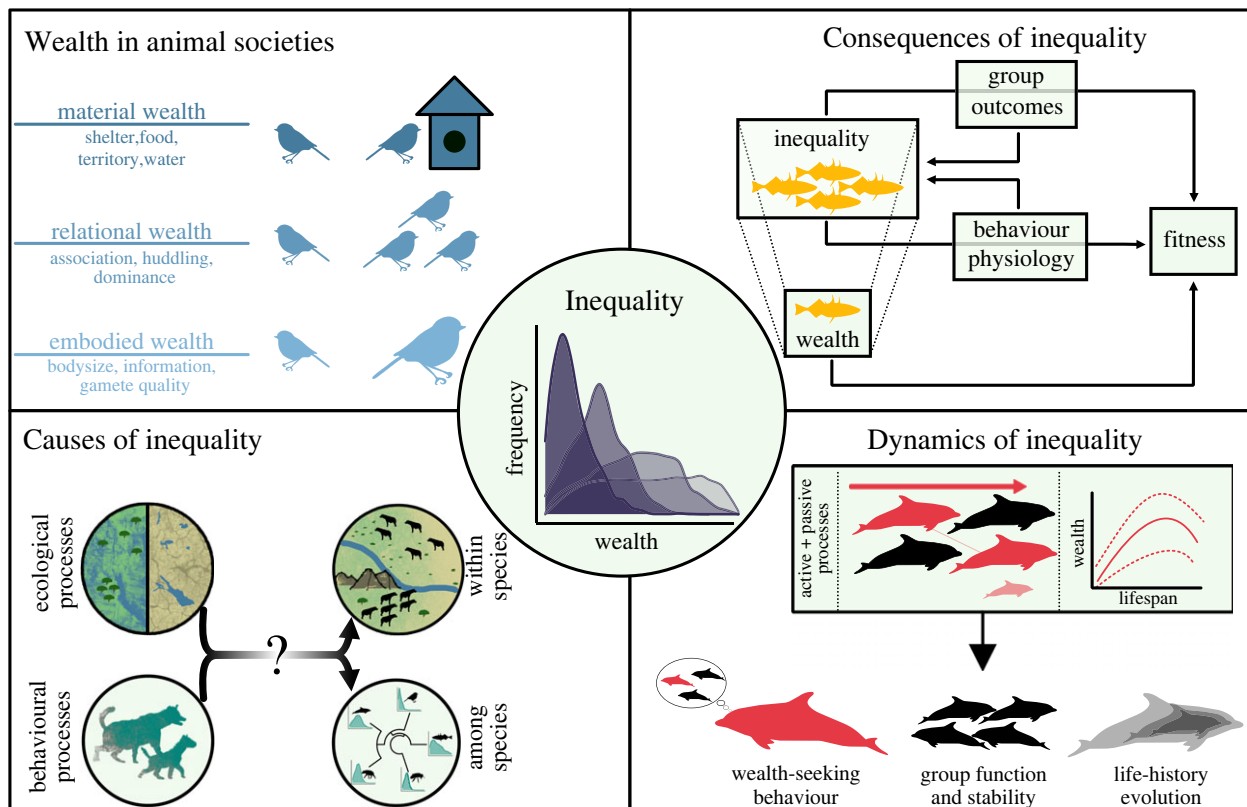

**Figure 1.** A schematic of the ecology of inequality. Centre circle: inequality describes the distribution of wealth among individuals, which can be measured using metrics borrowed from economics (box 1). Top left: wealth is taxonomically broad and occurs in many currencies, grouped into three aspects. Top right: inequality emerges from individual wealth through bottom-up causation and has a top-down influence on individual outcomes, both directly and via its effects on group outcomes. These effects are independent of the effects of wealth, but can feed back to influence wealth and inequality. Bottom left: multiple ecological (e.g. food/water distribution) and behavioural (e.g. wealth inheritance) processes are hypothesized to influence the amount of inequality in societies, but it is less clear at what scale this influence occurs or to what degree these processes operate across species. Bottom right: inequality is dynamic. Active and passive processes produce changes in wealth within an individual's lifetime and across generations, leading to typical wealth trajectories over the lifespan. The amount, timing and direction of wealth trajectories are expected to exert selection on individuals to optimize their experienced costs and benefits of sociality. (Online version in colour.)

---

**Box 1.** Measuring inequality.

Here, we provide a brief introduction to the methods for measuring inequality, intended to introduce the reader to what is an extensive body of literature in economics. Distributions can differ from pure equality in numerous ways [32–35]. When empirical wealth distributions are well described by the functional form of one or more distributions, inequality can be described analytically via the parameters specifying the distribution [36]. Alternatively, inequality can be measured by summarizing the amount of wealth held by individuals in a certain quantile (e.g. the proportion of total wealth held by the wealthiest 10% [37]) or by comparing the wealth of individuals in different quantiles. Finally, 'index' approaches summarize inequality into a single numerical index. The Gini index is the most commonly used metric of inequality, and although most often applied to income, it has also been used to study inequality in distributions of monetary wealth [38], land ownership [23], faculty production by universities [39], body size [40], plant sizes [41] and hermit crab shell sizes [25]. Because a single parameter cannot fully summarize the shape of a distribution, different indices are sensitive to different features of unequal distributions, so caution is warranted when indices disagree [32]. Finally, it is important to note that most of these methods were developed to describe inequality in large nation-states, and methodological challenges remain to facilitate comparative approaches to inequality in smaller societies such as those found in non-human systems [34,35,42,43].

---

of each aspect of wealth matter, and understanding the structural properties of wealth inequality is key to evolution. For example, niche construction may play a key role in evolution only when the intergenerational transmission of material wealth fundamentally alters how fitness is related to embodied aspects of wealth.

## 3. What are the consequences of inequality?

Inequality can influence outcomes for individuals directly or by impacting group outcomes (figure 1, top right). There is a

long history of sociological research describing different types of effects of wealth inequality (reviewed in [49]). Most directly, variation in individual wealth may translate into variation in outcomes, and such effects may be linear or non-linear. From an evolutionary ecology perspective, simple effects of wealth on fitness represent selection on various aspects of wealth, such as traits (embodied wealth), resource acquisition and defence (material wealth), or social behaviour (relational wealth). However, sociological approaches to wealth inequality also reveal other effects that may be

relevant to non-human societies. On top of simple wealth effects on outcomes, individuals are influenced by inequality in the distribution of wealth such that two equally wealthy individuals living in societies with different levels of wealth inequality might experience divergent outcomes. Here, we highlight three such effects: (i) the overall level of inequality at the group or society level may have effects beyond an individual's wealth; (ii) behavioural responses to inequality, and (iii) effects of inequality on group persistence or collective action.

Wealth and wealth inequality impact individual health and well-being [28,50–52]. In humans, more unequal societies are often associated with negative individual and societal outcomes [53,54]. An evolutionary comparison across primates, including humans, reveals that life-expectancy increases with lifespan equality, further indicating that inequality covaries with individual outcomes [55]. Inequality negatively impacts health and well-being through behavioural changes [56] or psychosocial stress [57]. In humans, inequality-induced stress is more extreme in societies that are more unequal, even for individuals of high social status [58]. Status-induced stress can affect both low- and high-wealth individuals, and who experiences most stress can depend on the dynamics of the social system [51,59,60]. Overall, widespread association between wealth inequality and individual outcomes supports the hypothesis that living in the context of wealth inequality is a 'fundamental cause' of a suite of negative outcomes [28,56,61].

Individuals attend to inequality within their societies and alter their behaviours accordingly. Experiments in primates, corvids and domestic dogs suggest that the perceived value of a resource is influenced by an individual's observations of the value of the resources their group-mates receive [62]. Individuals often then alter their social behaviour, for example by punishing individuals that receive the higher valued resource [63]. Similarly, subordinate queens of *Polistes fuscatus* wasps greatly increase aggression towards dominants when they perceive that dominants are claiming too unequal a share of reproduction [64]. In humans, an individual's wealth influences their perceptions about the degree of inequality in society [65] and their status-seeking behaviour [66]. In many species, individuals use social information about their status relative to their competitors when making decisions about how and with whom to compete [67]. In sum, intra-group competition and inequality are linked by a feedback loop involving individual perception of their own social status, the social status of others and the amount of inequality in the group. To understand this feedback loop, we should continue to explore how individuals perceive inequality, and how their response to inequality affects social structure. Systems where signals of wealth can be manipulated independently of actual wealth provide a means to experimentally manipulate perceived inequality.

Inequality can influence group outcomes such as group persistence and collective action. Reproductive skew theory [10,11] addresses how inequality in reproduction can affect the productivity or persistence of the group. Inequality can also influence a group's ability to cooperate or achieve collective action. In cooperation experiments with chimpanzees (*Pan troglodytes*), bonobos (*Pan paniscus*) and cotton-top tamarins (*Saguinus oedipus*), evidence suggests that species that divide the rewards of cooperation more equally are more likely to show cooperative behaviour [68,69]. Theoretical and empirical studies of collective action problems (e.g. public goods game) suggest that inequality has complex and often unpredictable effects on cooperative behaviour [70–77]. However, a rough pattern emerges in the literature suggesting that the effect of inequality on cooperation might depend on the type of wealth under consideration. In studies where individuals vary in the resources they can invest in cooperation (i.e. material wealth), inequality typically reduces cooperation [70–72]. However, inequality in social influence can promote cooperation by eliminating free-riders and overcoming coordination challenges [73–77]. Other evidence suggests that inequality can influence group outcomes by improving or impeding the function of groups, for instance by altering costs of coordination, resilience to variable environmental conditions, or ability to compete with other groups [73,75,78,79]. For example, burying beetles (*Nicrophorus nepalensis*) invest more in cooperation in the face of interspecific competitors [80]. A complex relationship between inequality and environment may explain global patterns in the evolution of cooperation: in both *Polistes* wasps and cooperatively breeding birds, the evolution of cooperative groups is associated with the environmental conditions that may increase the need for collective action (e.g. unpredictable environments: [81–83]). Overall, the complex results from theoretical studies suggest a need for empirical work on the links between inequality, individual outcomes and group function in animal systems.

## 4. What are the causes of inequality?

Multiple behavioural and ecological processes have been hypothesized to influence the amount of wealth inequality within societies, but the extent to which these mechanisms explain variation within versus among species is not fully clear (figure 1, bottom left). Some aspects of inequality seem to be relatively flexible, whereas others are more constrained. For example, in a population of olive baboons (*Papio anubis*) in Kenya, a mass mortality event prompted a long-term shift towards a more tolerant society with more equally distributed stress burdens, perhaps as a result of the death of the individuals that competed most intensely for high status [84]. However, a comparative network motif analysis of dominance hierarchies across many species suggests strong constraints on their structure related to transitivity of dominance relations [85]. Furthermore, in macaques, a suite of behaviours related to inequality in within-group conflict covary across species, producing macaque societies with different 'social styles' and suggesting potential phylogenetic constraints on wealth inequality [86,87]. More longitudinal and phylogenetic studies will be crucial to advance our understanding of plasticity and constraint in inequality across species.

What behavioural and ecological mechanisms influence variation in inequality within and among species? Ecological conditions—such as the patchiness, density and defensibility of resources—have long been hypothesized as a driver of material wealth inequality [1,2,9,88] (but see [89,90]). Additionally, inequality may be influenced by behavioural traits such as levelling coalitions used to control would-be dominants [91], aversion to unequal payoffs [62], preferences regarding perceived inequality [92], status-seeking behaviour [93], visibility of wealth [94] and cognitive processes relating to social competition [67]. Individuals can actively suppress

the wealth of others, as is seen in growth suppression by many fish [95] or the interruption of social bond formation in ravens (*Corvus corax*) [96], or subordinates may voluntarily reduce their own wealth to avoid conflict with group members [97]. Self-reinforcing dynamics—where 'rich-get-richer' feedbacks lead wealthy individuals to gain more wealth—can also influence the amount of inequality in societies [98] (see §5). Finally, these behavioural and ecological mechanisms interact. For example, the evolution of male coalitions in primates is explained by resource defensibility [99], and in vulturine guineafowl (*Acryllium vulturinum*), monopolization of clumped resources by dominants can lead to more egalitarian group movement decision-making [13].

Although drivers of inequality may differ among species or wealth aspects, some hypothesized causes of inequality are expected to operate across contexts. For example, the social transfer of wealth is one hypothesized driver of inequality that is likely to operate widely [3,4,16]. In a broad survey of human societies with diverse production systems, the increased fidelity of intergenerational transmission of wealth was associated with more extreme inequality [4,22]. In non-human animals, social inheritance of territory [100,101], knowledge [102,103], social relationships [46] and food caches [24] could provide ample contexts in which to test this hypothesis in diverse systems [16]. For instance, the social inheritance of dominance status in spotted hyenas and Old-World primates may drive inequality in dominance among lineages [29]. In fact, the widespread transmission of wealth across generations points to the evolutionary importance of non-genetic inheritance [45] and selection in response to multigenerational processes [104]. Another broadly operating hypothesized driver of inequality is intergroup conflict. When unequal groups are more effective or willing competitors, selection for success in intergroup conflicts can lead to increased within-group inequality in influence during collective action [79,105,106], and these leaders can also use their influence to increase inequality in other dimensions of wealth [107]. Here there is potential for positive feedback when the individuals that benefit most from intergroup conflict are also effective initiators of these conflicts, as seen in humans and banded mongoose (*Mungos mungo*) [108,109]. Finally, environmental stressors arising from climate change are expected to impact many species, highlighting another potentially broadly acting driver of inequality that we need to better understand. Studying shared processes influencing inequality in diverse wealth currencies and species is key to understanding the evolution of inequality and its role in societies.

## 5. How does inequality change over time?

Inequality is dynamic: neither the level of inequality nor an individual's wealth is fixed, and both can change over short or long timescales (figure 1, bottom right). One avenue for understanding these dynamics is through the economic concept of *social mobility,* which describes the dynamics of wealth measured at the individual or lineage level. Aggregating these measures across members of a social group reveals the society-level tendency for individuals or lineages to gain or lose wealth over time, producing more rigid or fluid societies. By integrating over time, social mobility mediates the link between inequality measured at a given time point and the processes or outcomes occurring over individual lifetimes.

Social mobility can vary in the timescale at which it occurs and the processes by which it arises. Intra- and inter-generational mobility classify the generational scale at which mobility occurs. *Intragenerational mobility* describes the degree to which individual wealth changes, producing wealth trajectories over the lifespan. *Intergenerational mobility* refers to the change in wealth within lineages across generations and is the type of social mobility most often studied in humans [110–112]. Examining the correlation between parents' and offspring's wealth provides an empirical measure of the extent to which an individual's position in society is malleable versus predetermined [113]. Increasingly, researchers are expanding the study of intergenerational mobility to include multigenerational effects, such as the effects of grandparents or other more distant kin [114,115].

Processes influencing social mobility can be active or passive: *active mobility* occurs when an individual's wealth changes with respect to their group-mates by reversing the wealth-ordering of individuals, whereas *passive mobility* occurs as a result of demographic processes such as births and deaths [116]. These demographic processes frequently produce gradual changes that have direct and indirect effects on social structure by removing and replacing individuals and altering existing social relationships [117]. In some cases, demographic changes can push societies over tipping points, or precipitous shifts in social structure that can show hysteresis [118]. Revolutions [119], mass mortality [84,119,120], group fissions [121], the arrival or loss of certain individuals [122–124] and expulsions of group members [125] are examples of active and passive processes that could produce precipitous changes. For instance, social perturbation experiments in captive fish, primates and mice demonstrate how removal of high-status individuals can lead to rapid behavioural, physiological and cognitive changes in other individuals [122–124].

The long-term additive combination of social mobility produces *long-run inequality*, which describes equilibrium patterns of inequality around which a society fluctuates [37,126], assuming such an equilibrium state exists. Understanding where a society sits relative to its expected equilibrium state will require long-term studies in the order of multiple generations. In turn, such work creates opportunities for exploring the forces that lead societies to deviate from or return to their equilibria. This long-run perspective could help us understand when and why societies may have distinctively low social mobility, leading to 'durable' inequality [127], or inequality that persists across individuals, time or generations [1]. Durable inequality can give rise to social classes, where individuals of different classes form social networks with different structures, face different mortality sources and cope differently with stressful conditions [60,128,129]. One process producing durable inequality is self-reinforcing dynamics, where already wealthy individuals accrue disproportionately greater wealth [130–133]. Preferential attachment and 'rich-club effect' models of social relationships demonstrate how relational wealth can show such self-reinforcing dynamics [134,135]. Frequency-dependent or fluctuating selection may be a counterforce that inhibits the buildup of durable inequality by altering fitness landscapes [136].

Patterns of social mobility may influence the evolution of a wide suite of behavioural strategies such as tolerance and wealth-seeking behaviour, as well as life-history traits related to pace of life (figure 1, bottom right). When upward intra-generational mobility is achieved through active processes,

selection is expected to favour individuals that challenge their group-mates, whereas conflict avoidance and tolerance should be favoured in species where upward intragenerational mobility is achieved through passive processes (e.g. social queuing; [137]). Low intergenerational mobility is expected to amplify selection on traits related to intragenerational mobility, as any changes within a generation are likely to persist and influence future generations. This hypothesized selection driven by social mobility reflects ways in which patterns in the dynamics of social structure can feed back to influence the evolution of individual traits [138], including life-history traits.

Contrasting hypotheses about the influence of social mobility on the stability of social groups highlights potential tradeoffs in the evolution of social structure. On the one hand, some have suggested that upward social mobility is crucial for long-term group stability, as individuals are expected to leave societies where they have no opportunity for wealth acquisition [126]. This pattern of upward mobility is prominent in societies where individuals 'queue' for wealth, such as in long-tailed manakins (*Chiroxiphia linearis*) [139], where individuals move up the queue through passive processes (e.g. death of wealthier individuals) [137,139,140]. By contrast, overly frequent active mobility can cause social instability, which is associated with negative consequences for individuals and societies [51,141–143]. These contrasting perspectives emphasize the need for theoretical and empirical work that generates and tests hypotheses about the link between social mobility and the functioning of societies in diverse species.

# 6. Conclusion and future directions

A key question in ecology and evolution is how the structure of groups arises and impacts the individuals that compose them [138]. Inequality in the distribution of wealth—be it relational, material or embodied—is a group-level feature that is hypothesized to impact individual and group outcomes. Here we coalesce disparate studies of inequality in biological systems into a research framework addressing inequality across ecological and evolutionary contexts and identify three overarching research foci.

First, how does inequality impact individuals beyond the simple effects of individual wealth? Evidence suggests that individuals attend to the amount of inequality within their societies, and that inequality *per se* may have adverse effects for individuals. Here, theoretical work has outpaced empirical work, and examining the impacts of inequality on individual and group outcomes in non-human systems will be fruitful. Experimental studies of inequality in laboratory populations is a promising tool for disentangling the effects of inequality from the effects of wealth. The recent surge in work on social dimensions of health and lifespan in non-human animals promises to shed light on potential avenues by which inequality influences fitness [28].

A second broad aim of the ecology of inequality is to understand the forces that cause inequality, both in the short term and at evolutionary timescales. Some aspects of inequality can be plastic—even sensitive to the behaviour of a single individual—whereas other aspects of inequality are evolutionarily constrained. The interplay between behavioural processes and environmental conditions (e.g. resource scarcity and competition) fundamentally shapes wealth inequality. Biogeographical and phylogenetic approaches may be useful here for identifying ecological and evolutionary patterns in wealth inequality at a global scale. Finally, feedback loops operating across species and types of wealth might explain why inequality is such a common feature of societies across the animal kingdom.

Third, it is crucial to take a dynamical perspective on inequality to understand selection on individual traits, long-term patterns in inequality, and the stability and persistence of groups. Social mobility—or changes in wealth—can occur owing to various processes and at different timescales, leading to higher-order patterns in inequality among individuals and their descendants, such as social classes or family dynasties. However, very little is known about the existence or implications of these higher-order patterns in inequality in non-human systems. Long-term studies that track groups and their constituents over multiple generations are uniquely situated to address this knowledge gap. Furthermore, we call for theoretical models that explore how lifetime patterns of social mobility impact the evolution of life-history traits and wealth-seeking behaviour.

Inequality is a curiously widespread feature of societies. The framework presented here offers a way forward for exploring the causes of inequality, its impacts on individuals and its role in social evolution. The framework allows inequality to be understood in specific contexts while also providing a means for comparative insight and the identification of general features of inequality operating across species and dimensions of wealth. This approach at once strengthens biological and sociological fields by integrating perspectives and facilitating the exchange of ideas, paving the way for new insights into ecological and evolutionary forces impacting social organisms.

Data accessibility. This article has no additional data.

Authors' contributions. E.D.S.: conceptualization, writing—original draft, and writing—review and editing; D.S.: conceptualization, writing—original draft, and writing—review and editing.

Both authors gave final approval for publication and agreed to be held accountable for the work performed herein.

Conflict of interest declaration. We declare we have no competing interests.

Funding. Open access funding provided by the Max Planck Society.

This work was supported by the University of Nebraska-Lincoln Population Biology Program of Excellence, NSF Grant OIA 0939454 via 'BEACON: an NSF Center for the Study of Evolution in Action', and the Alexander von Humboldt Foundation.

Acknowledgements. Thanks to Monique Borgerhoff Mulder, Mauricio Cantor, Danai Papageorgiou, members of the UNL School of Biological Sciences Behaviour Group, three anonymous reviewers and the reviews editor, Innes Cuthill, for helpful comments on prior versions of this manuscript.

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
