## [Peer Review File · Proceedings of the Royal Society B: Biological Sciences]

Review History

RSPB-2021-1583.R0 (Original submission)

Review form: Reviewer 1

Recommendation

Reject – article is not of sufficient interest (we will consider a transfer to another journal)

Scientific importance: Is the manuscript an original and important contribution to its field?

Acceptable

General interest: Is the paper of sufficient general interest?

Acceptable

Quality of the paper: Is the overall quality of the paper suitable?

Marginal

Is the length of the paper justified?

Yes

Should the paper be seen by a specialist statistical reviewer?

No

Do you have any concerns about statistical analyses in this paper? If so, please specify them explicitly in your report.

No

It is a condition of publication that authors make their supporting data, code and materials available - either as supplementary material or hosted in an external repository. Please rate, if applicable, the supporting data on the following criteria.

Is it accessible?

N/A

Is it clear?

N/A

Is it adequate?

N/A

Do you have any ethical concerns with this paper?

No

Comments to the Author

See attached. (See Appendix A)

Review form: Reviewer 2

Recommendation

Accept with minor revision (please list in comments)

Scientific importance: Is the manuscript an original and important contribution to its field?

Excellent

General interest: Is the paper of sufficient general interest?

Excellent

Quality of the paper: Is the overall quality of the paper suitable?

Good

Is the length of the paper justified?

Yes

Should the paper be seen by a specialist statistical reviewer?

No

Do you have any concerns about statistical analyses in this paper? If so, please specify them explicitly in your report.

No

It is a condition of publication that authors make their supporting data, code and materials available - either as supplementary material or hosted in an external repository. Please rate, if applicable, the supporting data on the following criteria.

Is it accessible?

N/A

Is it clear?

N/A

Is it adequate?

N/A

Do you have any ethical concerns with this paper?

No

Comments to the Author

The authors review work on inequality, its origins, and its effects in animal societies. This is a large undertaking, but the authors do well to summarize relevant literature and to argue that modeling of effects of inequality, independent of individual status or absolute wealth, is rare in non-humans. The authors provide a useful framework for future empirical work and present some testable hypotheses.

Im quite supportive of the paper, and have a number of comments that may be helpful to the authors. See below:

Upon first reading, it seems misleading, in the abstract and early in the introduction, to claim that explicit study of inequality is largely missing from evolutionary biology and ecology. I'm thinking of models of hierarchy formation and reproductive skew, as well as all the work connecting hierarchy and inequality in social relationships to health and reproduction in various animal societies (literatures which you later cite). Much hinges on what you mean by "explicit study of inequality" so be more precise if you can, in the abstract and introduction, to head off reader critiques. What you mean is that there is minimal study of how inequality distribution affects individuals in non-human societies, independent of individuals' absolute or relative "wealth" position, right?

Lines 80-82: for a citation of human data on associations between reproduction and different forms of wealth (including status/relational wealth), see <https://www.pnas.org/content/113/39/10824.short>.

Line 119, 125-126: studies finding effects of inequality on health have been heavily debated, with critics arguing that many studies fail to fully control for absolute wealth effects or account for the ecological fallacy. Perhaps mention and cite multi-level studies that deal with these issues, e.g. <https://elifesciences.org/articles/59437>.

Lines 128-132: Status anxiety may impact health not just via physiological effects of stress but indirectly via behavioral changes, including increased risk-taking (<https://pubmed.ncbi.nlm.nih.gov/27149981/>) or discounting of the future (e.g. <https://pubmed.ncbi.nlm.nih.gov/28073390/>)

Lines 132-133: Highest-ranking individuals in some studies appear to experience as much stress as the lowest-ranking (e.g. <https://www.ncbi.nlm.nih.gov/pmc/articles/PMC3433837/>), and stress at the top may be most likely when hierarchies are unstable, as Sapolsky has argued.

Line 145-146: I don't know of evidence in non-humans that individuals use information about level of inequality to adjust competitive behavior. Rather, individuals will use observation of others' wins and losses to adjust their behavior towards those individuals. And it seems only a minority of animal clades show evidence of use of such social inference in directing aggression towards conspecifics: <https://www.pnas.org/content/118/10/e2022912118>. Be more specific on what evidence exists.

Lines 164-168: how about some examples? Examples can be used to show when inequality improves vs. impedes group functioning. For example, leader-follower differentiation may improve group movement decisions when leaders have specialized knowledge or greater coordination ability, such as in elephants. Policing has been argued to provide a public good in certain non-human societies (by improving group stability/connectivity), and greater

interindividual differences in competitiveness make policing more likely (<https://www.nature.com/articles/nature04326>). On the other hand, when inequality increases/facilitates within-group competition, this can spur reduced investments in public goods: <https://academic.oup.com/beheco/article/23/4/735/221874>. This study argues that environmental risks can incentivize less within-group competition and greater public good provision, particularly by lower-ranking individuals: <https://royalsocietypublishing.org/doi/abs/10.1098/rspb.2020.1720>.

Line 179: the heading isn't numbered, which should shift up the numbering of the next two sections

Lines 197-206: I'd emphasize that individual behavioral traits interact with the kinds of ecological conditions you described. For example, human egalitarianism is maintained in part by leveling coalitions, but these are more likely to operate effectively in the absence of monopolizable wealth. Self-aggrandizing, status-seeking behavior is also curtailed when resources are riskily acquired, fostering greater cooperation and norms of humility. See this model: <https://pubmed.ncbi.nlm.nih.gov/33649461/>. With regard to the Matthew effect ("rich get richer") this too depends on extent of cooperation in a group, particularly the effect of cooperation on diffusion of social status through a group's network: <https://royalsocietypublishing.org/doi/full/10.1098/rspb.2019.1367>

Lines 218-220: Be more explicit here: the references here refer to within-group inequality in terms of decision-making ability in the context of collective action, right? Once such decision-making hierarchy emerges, it can be more likely to generate inequality in material wealth: <https://royalsocietypublishing.org/doi/full/10.1098/rspb.2014.1349>

Lines 254-256: An example here too would be useful- to show how demographic change produces change in mobility. You cite Scheidel: did the Black Death not just lower inequality but also increase mobility?

Lines 265-267: differentiated social networks may precede the emergence of social classes. Greater social connectivity in networks helps maintain egalitarianism (<https://royalsocietypublishing.org/doi/full/10.1098/rspb.2019.1367>) and greater deviation from panmixia in networks can foster emergence of social classes: <https://academic.oup.com/beheco/article/25/1/58/222376>

Lines 279-281: but wouldn't a faster pace-of-life erode the benefits of competing to be upwardly mobile? One solution is to make behavioral traits associated with such faster pace-of-life facultative, as evident in humans: forms of risk-taking (<https://pubmed.ncbi.nlm.nih.gov/27149981/>) and discounting of the future (e.g. <https://pubmed.ncbi.nlm.nih.gov/28073390/>) covary with SES.

Review form: Reviewer 3

Recommendation

Accept with minor revision (please list in comments)

Scientific importance: Is the manuscript an original and important contribution to its field?

Excellent

General interest: Is the paper of sufficient general interest?

Excellent

Quality of the paper: Is the overall quality of the paper suitable?

Excellent

Is the length of the paper justified?

Yes

Should the paper be seen by a specialist statistical reviewer?

No

Do you have any concerns about statistical analyses in this paper? If so, please specify them explicitly in your report.

No

It is a condition of publication that authors make their supporting data, code and materials available - either as supplementary material or hosted in an external repository. Please rate, if applicable, the supporting data on the following criteria.

Is it accessible?

N/A

Is it clear?

N/A

Is it adequate?

N/A

Do you have any ethical concerns with this paper?

No

Comments to the Author

Please see attached file. (See Appendix B)

Decision letter (RSPB-2021-1583.R0)

11-Aug-2021

Dear Dr Strauss:

I realise that this will be a source of frustration to you, but I am afraid that I cannot accept it for publication in its current state. The three reviewers reach very different conclusions -- two recommending acceptance with minor revisions and one recommending outright rejection -- but their comments are actually very similar. At a very fundamental level, your thesis that 'wealth' inequality has been understudied in non-human animals is wrong. That said, referees 1 and 3 found your article stimulating to read and the mere fact of being forced to think about possible parallels between studies in the human and evolutionary social societies was useful. I agree, but by no stretch of the imagination do I think that the necessary revisions are 'minor'. Aside from a proper acknowledgement of the many studies of causes and consequences of resource inequality in behavioural and evolutionary ecology, work needs to be done on a parity of treatment of examples -- as referee 3 points out, fitness measures should be treated as a consequence of resource inequality in non-human animal studies, and there needs to be consideration of fitness-related inequalities in humans. Another very important point - from referee 2 -- that needs addressing is clarity on how your review might actually change biologists' research questions and methods. I hope referee 2 forgives me if I'm wrong, but I suspect their negative view comes from

thinking that you might just be relabelling topics that evolutionary biologists are already studying, and those topics that cannot be relabelled are ones that are not found in non-humans because exist because of factors that are unique to our species. I think if you sit down and work through the referees' points systematically, and are able to incorporate them, there is a chance of a really stimulating review - for both evolutionary biologists and human social scientists. However, it will be a lot of work.

Please find below the comments received from the referees concerning your manuscript, not including confidential reports to the Editor. I hope you find these useful when considering whether to accept the challenge of revision. However please note that the offer of considering a revised ms is not a provisional acceptance.

- 1) A 'response to referees' document including details of how you have responded to the comments, and the adjustments you have made.
- 2) A clean copy of the manuscript and one with 'tracked changes' indicating your 'response to referees' comments document.
- 3) Line numbers in your main document.
- 4) Please read our data sharing policies to ensure that you meet our requirements <https://royalsociety.org/journals/authors/author-guidelines/#data>.

Best wishes,
Innes Cuthill

Prof. Innes Cuthill
Reviews Editor, Proceedings B
mailto: proceedingsb@royalsociety.org

Reviewer(s)' Comments to Author:
Referee: 1
Comments to the Author(s)
See attached.

Referee: 2

Comments to the Author(s)

The authors review work on inequality, its origins, and its effects in animal societies. This is a large undertaking, but the authors do well to summarize relevant literature and to argue that modeling of effects of inequality, independent of individual status or absolute wealth, is rare in non-humans. The authors provide a useful framework for future empirical work and present some testable hypotheses.

Im quite supportive of the paper, and have a number of comments that may be helpful to the authors. See below:

Upon first reading, it seems misleading, in the abstract and early in the introduction, to claim that explicit study of inequality is largely missing from evolutionary biology and ecology. I'm thinking of models of hierarchy formation and reproductive skew, as well as all the work connecting hierarchy and inequality in social relationships to health and reproduction in various animal societies (literatures which you later cite). Much hinges on what you mean by "explicit study of inequality" so be more precise if you can, in the abstract and introduction, to head off reader critiques. What you mean is that there is minimal study of how inequality distribution affects individuals in non-human societies, independent of individuals' absolute or relative "wealth" position, right?

Lines 80-82: for a citation of human data on associations between reproduction and different forms of wealth (including status/relational wealth), see <https://www.pnas.org/content/113/39/10824.short>.

Line 119, 125-126: studies finding effects of inequality on health have been heavily debated, with critics arguing that many studies fail to fully control for absolute wealth effects or account for the ecological fallacy. Perhaps mention and cite multi-level studies that deal with these issues, e.g. <https://elifesciences.org/articles/59437>.

Lines 128-132: Status anxiety may impact health not just via physiological effects of stress but indirectly via behavioral changes, including increased risk-taking (<https://pubmed.ncbi.nlm.nih.gov/27149981/>) or discounting of the future (e.g. <https://pubmed.ncbi.nlm.nih.gov/28073390/>)

Lines 132-133: Highest-ranking individuals in some studies appear to experience as much stress as the lowest-ranking (e.g. <https://www.ncbi.nlm.nih.gov/pmc/articles/PMC3433837/>), and stress at the top may be most likely when hierarchies are unstable, as Sapolsky has argued.

Line 145-146: I don't know of evidence in non-humans that individuals use information about level of inequality to adjust competitive behavior. Rather, individuals will use observation of others' wins and losses to adjust their behavior towards those individuals. And it seems only a minority of animal clades show evidence of use of such social inference in directing aggression towards conspecifics: <https://www.pnas.org/content/118/10/e2022912118>. Be more specific on what evidence exists.

Lines 164-168: how about some examples? Examples can be used to show when inequality improves vs. impedes group functioning. For example, leader-follower differentiation may improve group movement decisions when leaders have specialized knowledge or greater coordination ability, such as in elephants. Policing has been argued to provide a public good in certain non-human societies (by improving group stability/connectivity), and greater interindividual differences in competitiveness make policing more likely (<https://www.nature.com/articles/nature04326>). On the other hand, when inequality increases/facilitates within-group competition, this can spur reduced investments in public goods: <https://academic.oup.com/beheco/article/23/4/735/221874>. This study argues that environmental risks can incentivize less within-group competition and greater public good provision, particularly by lower-ranking individuals: <https://royalsocietypublishing.org/doi/abs/10.1098/rspb.2020.1720>.

Line 179: the heading isn't numbered, which should shift up the numbering of the next two sections

Lines 197-206: I'd emphasize that individual behavioral traits interact with the kinds of ecological conditions you described. For example, human egalitarianism is maintained in part by leveling

coalitions, but these are more likely to operate effectively in the absence of monopolizable wealth. Self-aggrandizing, status-seeking behavior is also curtailed when resources are riskily acquired, fostering greater cooperation and norms of humility. See this model:

<https://pubmed.ncbi.nlm.nih.gov/33649461/>. With regard to the Matthew effect (“rich get richer”) this too depends on extent of cooperation in a group, particularly the effect of cooperation on diffusion of social status through a group’s network:
<https://royalsocietypublishing.org/doi/full/10.1098/rspb.2019.1367>

Lines 218-220: Be more explicit here: the references here refer to within-group inequality in terms of decision-making ability in the context of collective action, right? Once such decision-making hierarchy emerges, it can be more likely to generate inequality in material wealth:
<https://royalsocietypublishing.org/doi/full/10.1098/rspb.2014.1349>

Lines 254-256: An example here too would be useful- to show how demographic change produces change in mobility. You cite Scheidel: did the Black Death not just lower inequality but also increase mobility?

Lines 265-267: differentiated social networks may precede the emergence of social classes. Greater social connectivity in networks helps maintain egalitarianism (<https://royalsocietypublishing.org/doi/full/10.1098/rspb.2019.1367>) and greater deviation from panmixia in networks can foster emergence of social classes:
<https://academic.oup.com/beheco/article/25/1/58/222376>

Lines 279-281: but wouldn’t a faster pace-of-life erode the benefits of competing to be upwardly mobile? One solution is to make behavioral traits associated with such faster pace-of-life facultative, as evident in humans: forms of risk-taking (<https://pubmed.ncbi.nlm.nih.gov/27149981/>) and discounting of the future (e.g. <https://pubmed.ncbi.nlm.nih.gov/28073390/>) covary with SES.

Referee: 3

Comments to the Author(s)

Please see attached file

Author's Response to Decision Letter for (RSPB-2021-1583.R0)

See Appendix C.

RSPB-2022-0500.R0

Review form: Reviewer 1

Recommendation

Accept with minor revision (please list in comments)

Scientific importance: Is the manuscript an original and important contribution to its field?

Good

General interest: Is the paper of sufficient general interest?

Excellent

Quality of the paper: Is the overall quality of the paper suitable?

Excellent

Is the length of the paper justified?

Yes

Should the paper be seen by a specialist statistical reviewer?

No

Do you have any concerns about statistical analyses in this paper? If so, please specify them explicitly in your report.

No

It is a condition of publication that authors make their supporting data, code and materials available - either as supplementary material or hosted in an external repository. Please rate, if applicable, the supporting data on the following criteria.

Is it accessible?

N/A

Is it clear?

N/A

Is it adequate?

N/A

Do you have any ethical concerns with this paper?

No

Comments to the Author

I appreciate the effort that the authors took to substantially re-write and re-organize the manuscript. I find this revised manuscript stimulating and can potentially draw the attention of ecologists and evolutionary biologists interested in animal societies to think more broadly and cohesively about inter-individual differences in niche, social relations and phenotypic values under one framework, such as the proposed 'wealth' framework that has been well developed in the research of human societies. I am still struggling a little bit with how operational this proposed wealth framework will be. For example, what would be the appropriate currencies (e.g. energy acquired in optimal foraging theory, inclusive fitness in cooperative breeding) that we could use to measure across different wealth dimensions (material, relational, embodied) and across study systems? That being said, I do agree that there is a need to develop a cohesive framework for studying and understating the interplay between inter-individual differences and animal societies.

Below I provide some suggestions that should help improve the clarity of this review:

1. The ecological and evolutionary unit of wealth inequality is necessarily dependent on the study system and research question. Therefore several different units are mentioned throughout the manuscript, such as population (demographic or genetic), group (as in social animals, which is often based on kinship), society (mostly used in human studies), community (appeared only in one sentence on page 8). While the usage of these related but not identical concepts appears to be appropriate, I suggest adding a glossary section to briefly define each of these units for which the "degree of inequality" can be measured in different contexts. This may help reduce potential confusions across readers from different research fields. For instance, a human society is ecologically closer to an animal population rather than an animal group (which may be closer to an extended family in human?).

2. On a related note, I advise not to include individual as a potential level for measuring inequality. Specifically on page 8: "Inequality at multiple levels (i.e., overall level of inequality of a community, as well as individual's relative position within the community...)", which I think is a bit confusing as inequality is really a measure of the distribution of wealth or fitness across some numbers of individuals (a group, a population, a society; Fig.1, page 7). At individual level, it is a "relative wealth/fitness position" that is the relevant measure here. Because the concept of inequality is logically a positive value (uniform distribution of wealth/fitness gives the minimum inequality = 0, which can only go up from there), it is difficult to apply a measure of inequality at individual level (do we say that an individual with a lot more wealth or a lot less wealth than other members of the group experiences the same amount of inequality?). I think "relative wealth/fitness position" is a fundamentally different measure from inequality as it carries both a sign (wealthier or poorer) and a magnitude (how much more or how much less).

3. Temporal changes in wealth inequality (social mobility) surely is an interesting aspect of wealth inequality. However, when concerned with evolutionary patterns (e.g. maintenance of wealth seeking behavior) or long-term ecological dynamics (e.g. stability of animal society), should the degree of social mobility be treated as a component of inequality measure (a population with higher social mobility is on average of lower inequality)? A bit clarification would be nice.

4. Are there also spatial dynamics of social mobility? As the authors pointed out, individuals may choose to leave a group of low temporal social mobility and join another group of higher temporal social mobility, which is well known in human societies. Therefore I would think social mobility involves spatiotemporal dynamics.

5. Three dimensions of wealth (material, relational, embodied) were defined in this manuscript and the concept of multidimensional wealth briefly mentioned (e.g. pages 6, 16). However, it was not clear whether the authors would recommend an approach to quantify inequality in multidimensional wealth space (similar to the Hutchinson's hypervolume; that is, a population, a society and/or a group may have one overall value of inequality corresponding to a volume in the 3-dimensional wealth space), or to treat these dimensions as separate but linked components (e.g. the amount of inequality in embodied wealth may lead to or covary with that in material wealth)? Note that "multidimensional" does not simply mean "considering several dimensions," and therefore it is better not to use these two terms as synonyms. It would be helpful to see some specific scenarios where a truly multidimensional approach will be appropriate and beneficial in studying wealth inequality.

6. I urge the authors to consider including a brief discussion on the possibility that neutral processes such as patchy resource distributions or phenotypic variation due to genetic drift can also lead to wealth inequality (e.g. patchy resources allowing some individuals to have more material wealth than others by chance, genetic drift allowing some individuals to have a larger body size than others by chance). Therefore a null model approach (i.e., comparing the observed amount of wealth inequality against the amount of inequality arose by chance) that is commonly employed in ecological and evolutionary research may also apply here.

Decision letter (RSPB-2022-0500.R0)

18-Mar-2022

Dear Dr Strauss

I am pleased to inform you that your manuscript RSPB-2022-0500 entitled "The ecology of wealth inequality in animal societies" has been provisionally accepted for publication in Proceedings B.

The referee is pleased with the effort you have gone to to take on board previous comments, as am I, but the referee does also suggest some further revisions to your manuscript which I think would be in your interest. It should not take you long. Therefore, I invite you to respond to the

referee's comments and revise your manuscript. Because the schedule for publication is very tight, it is a condition of publication that you submit the revised version of your manuscript within 7 days. If you do not think you will be able to meet this date please let us know.

When submitting your revision please upload a file under "Response to Referees" - in the "File Upload" section. This should document, point by point, how you have responded to the reviewers' and Editors' comments, and the adjustments you have made to the manuscript. We also require a copy of the revised manuscript showing track changes to be uploaded.

4) Data accessibility section and data citation

It is a condition of publication that data supporting your paper are made available either in the electronic supplementary material. Authors must complete the 'data accessibility' section in the submission system. This should list the database and accession number for all data from the article that has been made publicly available, for instance:

NB. From April 1 2013, peer reviewed articles based on research funded wholly or partly by RCUK must include, if applicable, a statement on how the underlying research materials – such as data, samples or models – can be accessed.

[http://datadryad.org/submit?journalID=RSPB&manu=\(Document not available\)](http://datadryad.org/submit?journalID=RSPB&manu=(Document not available)) which

will take you to your unique entry in the Dryad repository. If you have already submitted your data to dryad you can make any necessary revisions to your dataset by following the above link.

Please include the Dryad DOI in the Data Accessibility section and reference in the paper's bibliography.

Please see our Data Sharing Policies (<https://royalsociety.org/journals/authors/author-guidelines/>).

6) A media summary: a short non-technical summary (up to 100 words) of the key findings/importance of your manuscript.

Best wishes,
Innes Cuthill

Professor Innes Cuthill
Reviews Editor, Proceedings B
mailto: proceedingsb@royalsociety.org

Reviewer(s)' Comments to Author:

Referee: 1

Comments to the Author(s).

I appreciate the effort that the authors took to substantially re-write and re-organize the manuscript. I find this revised manuscript stimulating and can potentially draw the attention of ecologists and evolutionary biologists interested in animal societies to think more broadly and cohesively about inter-individual differences in niche, social relations and phenotypic values under one framework, such as the proposed 'wealth' framework that has been well developed in the research of human societies. I am still struggling a little bit with how operational this proposed wealth framework will be. For example, what would be the appropriate currencies (e.g. energy acquired in optimal foraging theory, inclusive fitness in cooperative breeding) that we could use to measure across different wealth dimensions (material, relational, embodied) and across study systems? That being said, I do agree that there is a need to develop a cohesive framework for studying and understating the interplay between inter-individual differences and animal societies.

Below I provide some suggestions that should help improve the clarity of this review:

1. The ecological and evolutionary unit of wealth inequality is necessarily dependent on the study system and research question. Therefore several different units are mentioned throughout the manuscript, such as population (demographic or genetic), group (as in social animals, which is often based on kinship), society (mostly used in human studies), community (appeared only in one sentence on page 8). While the usage of these related but not identical concepts appears to be appropriate, I suggest adding a glossary section to briefly define each of these units for which the "degree of inequality" can be measured in different contexts. This may help reduce potential confusions across readers from different research fields. For instance, a human society is ecologically closer to an animal population rather than an animal group (which may be closer to an extended family in human?).
2. On a related note, I advise not to include individual as a potential level for measuring inequality. Specifically on page 8: "Inequality at multiple levels (i.e., overall level of inequality of

a community, as well as individual's relative position within the community...)", which I think is a bit confusing as inequality is really a measure of the distribution of wealth or fitness across some numbers of individuals (a group, a population, a society; Fig.1, page 7). At individual level, it is a "relative wealth/fitness position" that is the relevant measure here. Because the concept of inequality is logically a positive value (uniform distribution of wealth/fitness gives the minimum inequality = 0, which can only go up from there), it is difficult to apply a measure of inequality at individual level (do we say that an individual with a lot more wealth or a lot less wealth than other members of the group experiences the same amount of inequality?). I think "relative wealth/fitness position" is a fundamentally different measure from inequality as it carries both a sign (wealthier or poorer) and a magnitude (how much more or how much less).

3. Temporal changes in wealth inequality (social mobility) surely is an interesting aspect of wealth inequality. However, when concerned with evolutionary patterns (e.g. maintenance of wealth seeking behavior) or long-term ecological dynamics (e.g. stability of animal society), should the degree of social mobility be treated as a component of inequality measure (a population with higher social mobility is on average of lower inequality)? A bit clarification would be nice.

4. Are there also spatial dynamics of social mobility? As the authors pointed out, individuals may choose to leave a group of low temporal social mobility and join another group of higher temporal social mobility, which is well known in human societies. Therefore I would think social mobility involves spatiotemporal dynamics.

5. Three dimensions of wealth (material, relational, embodied) were defined in this manuscript and the concept of multidimensional wealth briefly mentioned (e.g. pages 6, 16). However, it was not clear whether the authors would recommend an approach to quantify inequality in multidimensional wealth space (similar to the Hutchinson's hypervolume; that is, a population, a society and/or a group may have one overall value of inequality corresponding to a volume in the 3-dimensional wealth space), or to treat these dimensions as separate but linked components (e.g. the amount of inequality in embodied wealth may lead to or covary with that in material wealth)? Note that "multidimensional" does not simply mean "considering several dimensions," and therefore it is better not to use these two terms as synonyms. It would be helpful to see some specific scenarios where a truly multidimensional approach will be appropriate and beneficial in studying wealth inequality.

6. I urge the authors to consider including a brief discussion on the possibility that neutral processes such as patchy resource distributions or phenotypic variation due to genetic drift can also lead to wealth inequality (e.g. patchy resources allowing some individuals to have more material wealth than others by chance, genetic drift allowing some individuals to have a larger body size than others by chance). Therefore a null model approach (i.e., comparing the observed amount of wealth inequality against the amount of inequality arose by chance) that is commonly employed in ecological and evolutionary research may also apply here.

Author's Response to Decision Letter for (RSPB-2022-0500.R0)

See Appendix D.

Decision letter (RSPB-2022-0500.R1)

23-Mar-2022

Dear Mr Strauss

I am pleased to inform you that your manuscript entitled "The ecology of wealth inequality in animal societies" has been accepted for publication in Proceedings B.

Data Accessibility section

Open Access

Paper charges

Sincerely,

Proceedings B

Appendix A

Review of RSPB-2021-1583 “The ecology of inequality in animal societies”

Summary

This is an interesting review on how inequality in wealth among individuals and groups can help researchers better understand the structure and stability of human and animal societies. The concepts of inequality and wealth, along with their causes and consequences, are mainly developed for human societies, which was nicely reviewed in this manuscript. Based on the extensive knowledge and tool sets that we have gained from studying human societies, the authors proposed a framework that could be applied to animal societies, and illustrated possible applications using many examples of social animals, particularly non-human primates. The authors pointed out three broad sets of questions that could benefit from the framework of wealth and inequality: 1) how inequality in wealth affects individual fitness, both directly and indirectly (e.g. status stress)? 2) what are the ecological and evolutionary causes of inequality in wealth? 3) what are the evolutionary implications of inequality in wealth (e.g. pace-of-life, strength of sociality). While I applaud the efforts the authors took to make this connection between studies on human and animal societies, I found this review is lacking in details and depth to generate truly novel insights and practical guidance.

Major comments

1. A more comprehensive overview of current knowledge about animal societies is necessary, not just limited to selected examples that could match well to human studies.
I found this manuscript failed to properly synthesize current state of knowledge and decades of research in animal societies by ecologists and evolutionary biologists. For example, kin and group selection are well recognized mechanisms underlying social structure, which should have been carefully treated in this manuscript as they are strongly linked to social mobility. Furthermore, predation risk is known to be an important selection agent in the evolution of animal sociality. However, these discussions are nearly non-existence in this manuscript. While I understand the focus of this work is on inequality in wealth, but to really help readers appreciate how, by incorporating this framework in the studies of animal societies, we can gain additional insights (from what is already known in ecological and evolutionary research), a more comprehensive overview of current knowledge about animal societies is necessary.
2. How does the wealth concept add to the existing concepts of niche, social niche and phenotypic traits in ecology and evolution?
The three dimensions of wealth, namely material, relational and embodied, are mapped to niche, social niche and phenotypic traits in ecology and evolution. However, if these three sets of concepts are perfectly mapped (equivalent), what is the benefit of replacing the terminology? Are there any fundamental differences between material wealth and niche, relational wealth and social niche, embodied wealth and phenotypic traits? For example, when I think about inequality in embodied wealth measured by body size, I would equate it to phenotypic variation in body size among individuals.

3. How does the concept of inequality add to the existing concepts of inter-individual difference or individual variability in ecology and evolution?
Inter-individual differences in niche use, social status, phenotypic trait are widely studied in ecology and if it has the same meaning as inequality as defined in this manuscript, I fail to see true novelty of the proposed framework. Furthermore, it seems to me that the center panel in Fig. 1 (inequality describes the distribution of wealth among individuals) is the same as niche use frequency or phenotypic distribution (by replacing wealth on the x-axis with niche or trait value).
4. Animal societies can range from eusocial insects to birds that engage in seasonal cooperative breeding. What types of animal societies this framework can apply and where is the limit (would this framework help studying eusocial animals)? Avoid overstating the significance and applicability of this framework as human is only one species (and a very unique one) and the variation among human societies is probably very small compared to that among animal societies.

Appendix B

Ms. Ref. No.: MEC-21-0683
Title: The ecology of inequality in animal societies
Authors: Eli Strauss & Daizaburo Shizuka

Overview and general recommendation:

The origin and impacts of inequality in human societies is widely studied, while often only studied in animal societies in terms of dominance or reproductive skew and has rarely been linked to economics, anthropology and psychology. The current paper is an interesting and comprehensive review of inequality in both human and animal societies and aims to integrate the two fields identifying common principles when appropriate.

The paper is structured around four main section which are clearly presented in Figure 1: a) What is wealth and inequality in animal societies? b) What are the consequences of inequality? Although not defined as such, there is a section on c) Why are societies unequal? Which could also be labelled What are the causes of inequality? to better fit with Figure 1. Finally, there is a section on d) How does inequality change over time?

The paper is well researched with an impressive 140 references that are relevant to each section. The paper successfully brings together the knowledge and methods used in human societies and in particular on the inequality of wealth, to draw parallels with animal societies. I am more familiar with the ecology and evolution literature and while most relevant references are cited, I suggest ways below in which they could be better discussed to highlight the work that has been carried out in animal societies. One point which is not explicitly discussed, is that in human societies research has concentrated on inequality in wealth itself, which is access to resources, physical condition and social connections, whereas research in animal societies has concentrated more on the consequences on inequality of wealth, such as inequality in reproductive success and ultimately fitness. I would suggest that this should be more apparent in the two different sections, the strengths of research in human vs animal societies and vice versa as a function of which traits have been measured. I disagree with the authors claim that “very little work in non-humans has explored pathways by which inequality impacts individuals, societies and evolution“ and I have suggested ways in which this could be modified. I have suggested multiple measures of inequality in reproductive success and even of wealth that have not been mentioned in this paper. Certain areas could be expanded upon including: perception of inequality in animal societies, the altering of social behaviours and conflict resolution.

Major comments:

2. What is wealth and inequality in animal societies? As stated on lines 20-21: Inequality in access to resources, physical condition and sociality (*measures of wealth*) translates into differences in health, longevity and reproductive success and ultimately fitness. Box 1 is mainly concerned with a description of different measures of human inequality in terms of wealth (*income*), but also land ownership, social connection, faculty production and body size. Box 1 also measures inequality in animal societies in

terms of reproductive success and body size. However, I would consider reproductive success in animal societies as the outcome of measures of wealth, whereas no fitness measures are described in Box 1 for humans (health, longevity and reproductive success). Reproductive success fits more into **3. What are the consequences of inequality?** Alternatively, Box 1 could be expanded to include measuring inequality in wealth AND measuring inequality in reproductive success (see below).

3. What are the consequences of inequality?

Lines 122-124: I would argue against that “very little work in non-humans has explored pathways by which inequality impacts individuals, societies and evolution” as the vast field of sexual selection in ecology and evolution describes and widely quantifies how “inequality in wealth” i.e. access to resources, physical condition and sociality (*measures of wealth in this paper*) translates into reproductive success. On lines 198-199, the classic paper by Emlen & Oring 1977 is referenced in relation to the behavioural and ecological conditions as drivers of inequality, which is accurate, but a lot of work stemming from this theory discusses the consequences of inequality on mating and social systems.

Classical sexual selection theory predicts that inequality will be higher (the higher the variance in reproductive and mating success) when the more the access to either one of the sexes, or to reproductive opportunities (*material wealth*), is limiting, the stronger will be the competition between individuals of the opposite sex (*embodied wealth*) and the stronger the sexual selection (*inequality*) (Emlen & Oring 1977; Andersson, 1994; Bateman, 1948; Darwin, 1871; Trivers, 1972). The potential for such multiple mating depends on feasibility within an individual’s time budget (little or no parental investment), and whether multiple mates, or resources required for multiple mating (*material wealth*), can be monopolised in time and space (Emlen & Oring 1977). The higher fitness gain from having multiple partners (*inequality*), the sexual selection or Bateman gradient (Bateman, 1947, Arnold & Wade, 1994), is the cause of sexual selection, giving rise to stronger male-male competition, female mate choice, and greater variation in structural and/or behavioral traits in males (*embodied wealth*). Sexual selection determines the resulting mating system (*society*) and explains its evolution.

In terms of measures of inequality of reproductive success, there are many. I have already mentioned the Bateman gradient, but there are also Bateman’s variances (the opportunity of sexual selection I_s and selection, I), the index of resource monopolization (Q) and the Morista index. Furthermore, other measures of sexual selection include selection differentials (s') and selection gradients (β') that measure the direct selection on phenotypic characters to reveal the target(s) of sexual selection (Lande & Arnold 1983). These coefficients quantify the intensity of sexual selection and have greater predictive value in relation to evolutionary change. A few papers have compared these different measures: Mills *et al.*, Proc Roy Soc Lond B (2007) 274, 143–150, as well as Fairbairn & Wilby (2001) and (Jones *et al.* 2004, 2005) referenced within.

Individual perception of inequality: Lines 140-146: There are also mechanisms in place that perceive inequality in animal societies. For example, males falsely signaling their reproductive quality (*embodied wealth*) will either suffer mortality due to the cost of maintaining the signal or injury/death after losing in competition to other males. Zahavi’s handicap principle posits that signals will provide reliable information about the quality

of signalers, provided that they are costly to produce (Zahavi 1975, 1977). This sexual selection principal has been proven both theoretically (Enquist 1985; Grafen 1990; Godfray 1991; Maynard-Smith 1991; Johnstone and Grafen 1992) and empirically (e.g., Andersson 1994; Johnstone 1995; Møller 1995; Mappes et al. 1996; Kilpimaa et al. 2004). The expectation of reliability is inherent in both “viability indicator” and Fisherian mechanisms of sexual selection (that are but a continuum of a single process, Kokko et al. 2002) manifest in sons as either superior survivorship and growth or attractiveness, respectively (Greenfield and Rodriguez 2004).

A female’s choice of mate may be based on a signal or other advertisement feature (*embodied wealth*), that is a reliable indicator of a potential mate’s phenotype, and ultimately their heritable fitness (Moore 1994; Welch et al. 1998; Møller and Alatalo 1999; Doty and Welch 2001). The simplest mechanism for the maintenance of signal reliability is physical constraint, such as the carotenoid-based plumage coloration in male house finches, *Carpodacus mexicanus*, which accurately indicates nutritional condition, and thus health or foraging ability (Hill and Montgomerie 1994) and females selecting brighter males acquire higher quality mates (Hill 1991). This is an example of a reliable signal.

Lines 142-151: An example of altering social behavior in animal societies can be found with subordinate *Polistes* paper wasps show increased aggression to the dominant queen if the subordinate eggs are removed – this represents an increase in reproductive skew with the subordinates receiving a decrease in reproduction and hence they retaliate. Reeve, H.K. & Nonacs, P. (1992) *Nature* 359, 823-825.

Inequality and group outcomes: Lines 152-170: There are also mechanisms in place in animal societies, to resolve conflicts of inequality. I am not familiar with the human society literature and hence whether there is an equivalent, but it would be interesting to include a section on conflict resolution. In some animal societies, for example in clownfish groups, there is inequality in reproduction with only the two largest individuals reproducing and yet smaller non-reproducing individuals stay within the group and there is no conflict. In this example, group conflict is resolved with the maintenance of size differences between individuals ensuring that smaller individuals, or subordinates, do not become a threat and challenge the reproductive status of the larger or dominant individuals. Here are two relevant publications:

Buston PM (2003) Size and growth modification in clownfish. *Nature* 424:145–146

Wong et al (2016) The four elements of within-group conflict in animal societies: an experimental test using the clown anemonefish, *Amphiprion percula*. *Behav Ecol Sociobiol* 70:1467–1475

To my knowledge, it is not yet known if the subordinates “pay to stay”, a mechanism by which subordinate individuals regulate their own growth so as not to incur eviction and remain queuing within the group. Subordinates “pay to stay” by which subordinate individuals increase cooperative effort, or whether their growth is under social control by the dominant individuals. These are two very different mechanisms and may be compared with cooperation and “manipulation” in humans.

If this is indeed similar to cooperation in humans, evidence of the “pay to stay” mechanism has also been reported in other fish and insects: in cichlids, *Neolamprologus*

pulcher (Heg et al., 2004; Brintjes & Taborsky, 2008), gobies *Paragobiodon xanthosomus*, (Wong et al., 2008) and paper wasps, *Polistes dominula* (Grinsted & Field, 2017). There are dominance hierarchies based on weight in many social mammals (Veiberg et al., 2004) and in other social mammals subordinate females can be aggressively evicted by older dominants (Kappeler & Fichtel, 2012; Pope, 2000). However, no mammalian studies have yet investigated whether individuals modify their growth rates or levels of cooperation to minimize conflict with the dominant.

Line 179: **Why are societies unequal?:** To follow the current structure of the paper, this section could be **4. What are the causes of inequality?** Also reference should be made to Figure 1 (bottom left).

Line 227: **4. How does inequality change over time?** Actually refers to **Dynamics of inequality** and reference should be made to Figure 1 (bottom right).

Minor comments:

A personal suggestion I would replace “like” with “such as” throughout: line 70, line 72, line 250, line 334.

Lines 71- 82: The order in Figure 1 and in the summary on Lines 68-71 is Material wealth, Relational wealth and then Embodied wealth. However on lines 71-87, this order is changed, a small point, but makes reading easier. On lines 71-73 you start with Embodied wealth, Lines 73-78 you shift to Material wealth and finally on lines 78-82 Relational wealth.

Lines 71: Embodied wealth in the text should include body size as it does in Figure 1, as well as ornament size, both visual and olfactory and courtship displays

Line 109: I would define WEIRD.

Line 104: typo, either “an individual” or “individuals”.

References

Some references have used a capital letter at the beginning of each word of the title and need to be corrected: references 4, 5, 6, 8, 11, 18, 25, 41, 48, 54, 59, 62, 71, 75, 85, 89, 96, 100, 106, 107, 108, 110, 121, 126, 127, 132 and 135.

All Science references have strange section in the title (**80-**). I have seen this before when using Mendeley. See references 6, 15, 31, 40, 50, 82, 94, 132, 134,

Reference 60: is missing journal specifics (number and page numbers)

Reference 121: Species name needs to be placed in italics, *Macaca mulatta*

Appendix C

Editor

I realise that this will be a source of frustration to you, but I am afraid that I cannot accept it for publication in its current state. The three reviewers reach very different conclusions -- two recommending acceptance with minor revisions and one recommending outright rejection -- but their comments are actually very similar. At a very fundamental level, your thesis that 'wealth' inequality has been understudied in non-human animals is wrong. That said, referees 1 and 3 found your article stimulating to read and the mere fact of being forced to think about possible parallels between studies in the human and evolutionary social societies was useful. I agree, but by no stretch of the imagination do I think that the necessary revisions are 'minor'.

We have given this some careful thought, reflecting on the supportive and critical comments from yourself and the reviewers. We agree that our originally stated thesis—that evolution and ecology is missing an explicit study of inequality in animals—is not quite right. Furthermore, as pointed out by reviewer 2, “explicit study of inequality” is a vague statement to begin with, and we think this vagueness lies at the heart of our initial overstatement of this project. To address these issues, we have rewritten our introduction to acknowledge the ways in which inequality is currently studied in animal societies. Our central thesis is now more specific, that the multidimensional wealth and inequality framework we introduce presents the opportunity to reintegrate key evolutionary concepts that have often been perceived as alternatives (e.g., trait evolution, niche construction, extended phenotypes) as different dimensions of the wealth-fitness link, allowing for new connections among these topics and with the literature on the causes and consequences of inequality originating from other fields (e.g., anthropology, economics). This new thesis clarifies the role of social systems as central to evolutionary biology, is closer to the heart of what the paper achieves, and avoids underselling what we agree is a robust literature on inequality in animals.

This thesis also specifies the scope of the paper, which focuses on applying the concepts of wealth and inequality to non-human animal systems and leaves questions about the evolution of inequality in humans for future work. We made this choice because it allows for a clearer and more in-depth discussion of inequality in non-human animals, and because considerable work on evolutionary approaches to inequality and social structure in humans already exists (e.g., Mattison et al. 2016, Haynie et al. 2021, Shennan et al. 2011, Borgerhoff Mulder 2009, Kaplan et al. 2009, Gintis et al. 2015). We now highlight this work in the introduction, but do not explore it in detail.

Aside from a proper acknowledgement of the many studies of causes and consequences of resource inequality in behavioural and evolutionary ecology, work needs to be done on a parity of treatment of examples -- as referee 3 points out, fitness measures should be treated as a consequence of resource inequality in non-human animal studies, and there needs to be consideration of fitness-related inequalities in humans.

We agree that we need to do more to clarify the relationship between resource/wealth inequality and fitness--i.e., that inequality in fitness is a consequence of wealth inequality. This is now discussed explicitly in the introduction. To be clearer about where we are talking about inequality in wealth vs. inequality in fitness, we have standardized language throughout the paper to refer to these two concepts respectively as “wealth inequality” and “fitness variation”.

After the introduction, our aim is to focus primarily on the evolutionary ecology of animal social systems. Thus, we only briefly mention fitness-related inequalities in humans, because that falls outside the scope of the paper.

Another very important point - from referee 2 -- that needs addressing is clarity on how your review might actually change biologists' research questions and methods. I hope referee 2 forgives me if I'm

wrong, but I suspect their negative view comes from thinking that you might just be relabelling topics that evolutionary biologists are already studying, and those topics that cannot be relabelled are ones that are not found in non-humans because exist because of factors that are unique to our species.

Our central thesis, which hopefully comes across more clearly, is that the “ecology of inequality” approach presents an opportunity to reintegrate concepts in evolutionary biology (e.g., trait evolution vs. niche construction, etc.), and it is also a way to clarify the role of social systems in shaping evolution. We are clear that some key concepts (e.g., the three main dimensions of wealth) are analogous to existing topics of interest to evolutionary biologists. But rather than considering this as simply “relabeling”, we argue that applying this existing framework of wealth from human-oriented literature helps us see in new ways how those topics are connected. Whether or not this perspective will actually change biologists’ questions and methods is perhaps out of our control, though we believe we offer some concrete ideas for methods and questions that could be pursued further.

I think if you sit down and work through the referees' points systematically, and are able to incorporate them, there is a chance of a really stimulating review - for both evolutionary biologists and human social scientists. However, it will be a lot of work.

Thank you and to the reviewers for these helpful comments and the constructive criticism.

Referee 1

Summary

This is an interesting review on how inequality in wealth among individuals and groups can help researchers better understand the structure and stability of human and animal societies. The concepts of inequality and wealth, along with their causes and consequences, are mainly developed for human societies, which was nicely reviewed in this manuscript. Based on the extensive knowledge and tool sets that we have gained from studying human societies, the authors proposed a framework that could be applied to animal societies, and illustrated possible applications using many examples of social animals, particularly non-human primates. The authors pointed out three broad sets of questions that could benefit from the framework of wealth and inequality: 1) how inequality in wealth affects individual fitness, both directly and indirectly (e.g. status stress)? 2) what are the ecological and evolutionary causes of inequality in wealth? 3) what are the evolutionary implications of inequality in wealth (e.g. pace-of-life, strength of sociality). While I applaud the efforts the authors took to make this connection between studies on human and animal societies, I found this review is lacking in details and depth to generate truly novel insights and practical guidance.

Thanks to the reviewer for their constructive criticism. We have taken some steps to clarify the primary contributions of the paper, and we hope that the novel insight and value of the paper is now communicated more clearly.

Major comments

1. A more comprehensive overview of current knowledge about animal societies is necessary, not just limited to selected examples that could match well to human studies. I found this manuscript failed to properly synthesize current state of knowledge and decades of research in animal societies by ecologists and evolutionary biologists. For example, kin and group selection are well recognized mechanisms underlying social structure, which should have been carefully treated in this manuscript as they are strongly linked to social mobility. Furthermore, predation risk is known to be an important selection agent in the evolution of animal sociality. However, these discussions are nearly

non existence in this manuscript. While I understand the focus of this work is on inequality in wealth, but to really help readers appreciate how, by incorporating this framework in the studies of animal societies, we can gain additional insights (from what is already known in ecological and evolutionary research), a more comprehensive overview of current knowledge about animal societies is necessary.

The reviewer is correct that we do not offer a comprehensive overview of current knowledge about the evolution of animal societies in this paper. However, this is by design, as the goal of the paper is to offer a new perspective on the role of inequality in animal societies, and we feel that dedicating more space to a review of topics that are well covered in other papers and books would take away from our unique contributions. The perspective presented here does not clash with well-established theories of kin or group selection, nor does it have much to do with the role of predation in social evolution. In lieu of a thorough review, we now include a new paragraph in the introduction that touches on various key concepts in social evolution, such as dominance hierarchies, social selection, reproductive skew, collective action, kin selection and multilevel selection. We also discuss the intersection between ecology and behavioral processes in the “causes of inequality” section. We hope this satisfies the desire to place this manuscript within the landscape of work on animal social evolution.

2. How does the wealth concept add to the existing concepts of niche, social niche and phenotypic traits in ecology and evolution? The three dimensions of wealth, namely material, relational and embodied, are mapped to niche, social niche and phenotypic traits in ecology and evolution. However, if these three sets of concepts are perfectly mapped (equivalent), what is the benefit of replacing the terminology?

Our goal was not to replace the terminology, but to illustrate the parallels between the conceptualizations of wealth--i.e., dimensions of inequality that can potentially affect fitness. We believe that illustrating these parallels accomplish two key things: (1) We show that these “hot-button” debates (e.g., is an “extended synthesis” that includes niche construction necessary?) explore how different dimensions (aspects) of wealth impact fitness. That is, it shows us a path towards consolidating these concepts and clarifying where distinctions between them are or are not necessary. (2) Having made the parallel between evolutionary concepts and conceptualizations of wealth, we illustrate how existing approaches from the human-oriented literature can be applied to understand wealth inequality in an ecological and evolutionary framework. We now state this explicitly in Section 2.

Are there any fundamental differences between material wealth and niche, relational wealth and social niche, embodied wealth and phenotypic traits? For example, when I think about inequality in embodied wealth measured by body size, I would equate it to phenotypic variation in body size among individuals.

We argue that these aspects of wealth accommodate these ecological concepts (niche, social niche, phenotypic trait), but may encompass other dimensions not captured with these terms. Taking “embodied wealth” for example: this includes phenotypic traits such as body size, but can also include personal information. Furthermore, viewing these as different dimensions of wealth leads to questions about the drivers of inequality in these domains, including the possibility that these types of inequality share some causes and consequences.

3. How does the concept of inequality add to the existing concepts of inter-individual difference or individual variability in ecology and evolution?

Inter-individual differences in niche use, social status, phenotypic trait are widely studied in ecology and if it has the same meaning as inequality as defined in this manuscript, I fail to see true novelty of the proposed framework.

Yes, wealth distribution is, fundamentally, the inter-individual variability in wealth. But we suggest that there are new opportunities for ecologists and evolutionary biologists by broadening our view on where such variability (inequality) comes from, how it affects individuals, and how it changes across time. For example, we illustrate that frameworks for understanding patterns of inequality exist (e.g., in economics). The section on consequences of inequality has now been edited to first acknowledge that an individual's wealth (i.e., their position in the unequal society) has direct effects on individual outcomes, but we then focus our discussion on various ways in which inequality per se (or the perception of inequality) also have effects. We also show how the interplay between ecology and social processes (intra- and inter-generational) are critical to understanding why inequality varies among societies. Our section on dynamics of inequality offer new views on how to integrate within- and across-generation processes of impact wealth inequality and how this might impact evolution. We try to highlight these main contributions in the conclusion.

Furthermore, it seems to me that the center panel in Fig. 1 (inequality describes the distribution of wealth among individuals) is the same as niche use frequency or phenotypic distribution (by replacing wealth on the x-axis with niche or trait value).

The intent here is not to simply replace “trait” or “niche use frequency” with “wealth”, but rather to illustrate how there are existing frameworks from economics to measure aspects of wealth from these patterns of distribution, as detailed in Box 1. We have edited the figure captions to clarify this.

4. Animal societies can range from eusocial insects to birds that engage in seasonal cooperative breeding. What types of animal societies this framework can apply and where is the limit (would this framework help studying eusocial animals)? Avoid overstating the significance and applicability of this framework as human is only one species (and a very unique one) and the variation among human societies is probably very small compared to that among animal societies.

We have broadened the taxonomic diversity of examples throughout the manuscript. We now touch on both eusocial and non-eusocial invertebrates as well as birds and mammals. We explicitly draw parallels between cooperative breeding in wasps and birds in the section on causes of inequality.

Referee: 2

Comments to the Author(s)

The authors review work on inequality, its origins, and its effects in animal societies. This is a large undertaking, but the authors do well to summarize relevant literature and to argue that modeling of effects of inequality, independent of individual status or absolute wealth, is rare in non-humans. The authors provide a useful framework for future empirical work and present some testable hypotheses.

Im quite supportive of the paper, and have a number of comments that may be helpful to the authors. See below:

Upon first reading, it seems misleading, in the abstract and early in the introduction, to claim that explicit study of inequality is largely missing from evolutionary biology and ecology. I'm thinking of models of hierarchy formation and reproductive skew, as well as all the work connecting hierarchy

and inequality in social relationships to health and reproduction in various animal societies (literatures which you later cite). Much hinges on what you mean by “explicit study of inequality” so be more precise if you can, in the abstract and introduction, to head off reader critiques. What you mean is that there is minimal study of how inequality distribution affects individuals in non-human societies, independent of individuals’ absolute or relative “wealth” position, right?

This is a very helpful comment that points out some lack of clarity in our setup of the paper. We now address the connections between hierarchies, reproductive skew, etc. in our revamped introduction.

Lines 80-82: for a citation of human data on associations between reproduction and different forms of wealth (including status/relational wealth), see <https://www.pnas.org/content/113/39/10824.short>.

Thanks for this recommendation, we now cite this paper here.

Line 119, 125-126: studies finding effects of inequality on health have been heavily debated, with critics arguing that many studies fail to fully control for absolute wealth effects or account for the ecological fallacy. Perhaps mention and cite multi-level studies that deal with these issues, e.g. <https://elifesciences.org/articles/59437>.

Thanks for bringing this impressive and highly relevant recent study to our attention. This paper is now cited in the manuscript.

Lines 128-132: Status anxiety may impact health not just via physiological effects of stress but indirectly via behavioral changes, including increased risk-taking (<https://pubmed.ncbi.nlm.nih.gov/27149981/>) or discounting of the future (e.g. <https://pubmed.ncbi.nlm.nih.gov/28073390/>)

This is a compelling and relevant argument, and we now cite Pepper & Nettle 2017 here.

Lines 132-133: Highest-ranking individuals in some studies appear to experience as much stress as the lowest-ranking (e.g. <https://www.ncbi.nlm.nih.gov/pmc/articles/PMC3433837/>), and stress at the top may be most likely when hierarchies are unstable, as Sapolsky has argued.

We agree this is a good point to include in this section, and we have added it.

Line 145-146: I don’t know of evidence in non-humans that individuals use information about level of inequality to adjust competitive behavior. Rather, individuals will use observation of others’ wins and losses to adjust their behavior towards those individuals. And it seems only a minority of animal clades show evidence of use of such social inference in directing aggression towards conspecifics: <https://www.pnas.org/content/118/10/e2022912118>. Be more specific on what evidence exists.

We were insufficiently clear about what social information we were referring to. We’ve reworded the sentence to read: “ In many species, individuals use social information about their status relative to their competitors when making decisions about how and with whom to compete.”

Lines 164-168: how about some examples? Examples can be used to show when inequality improves vs. impedes group functioning. For example, leader-follower differentiation may improve group movement decisions when leaders have specialized knowledge or greater coordination ability, such as in elephants. Policing has been argued to provide a public good in certain non-human societies (by improving group stability/connectivity), and greater interindividual differences in competitiveness

make policing more likely (<https://www.nature.com/articles/nature04326>). On the other hand, when inequality increases/facilitates within-group competition, this can spur reduced investments in public goods: <https://academic.oup.com/beheco/article/23/4/735/221874>. This study argues that environmental risks can incentivize less within-group competition and greater public good provision, particularly by lower-ranking individuals: <https://royalsocietypublishing.org/doi/abs/10.1098/rspb.2020.1720>.

Adding empirical examples here is a good idea. We have added the last citation the reviewer recommends here, as well as another example on the link between inequality, environment and evolution of cooperation.

Line 179: the heading isn't numbered, which should shift up the numbering of the next two sections

Fixed.

Lines 197-206: I'd emphasize that individual behavioral traits interact with the kinds of ecological conditions you described. For example, human egalitarianism is maintained in part by leveling coalitions, but these are more likely to operate effectively in the absence of monopolizable wealth. Self-aggrandizing, status-seeking behavior is also curtailed when resources are riskily acquired, fostering greater cooperation and norms of humility. See this model: <https://pubmed.ncbi.nlm.nih.gov/33649461/>. With regard to the Matthew effect ("rich get richer") this too depends on extent of cooperation in a group, particularly the effect of cooperation on diffusion of social status through a group's network: <https://royalsocietypublishing.org/doi/full/10.1098/rspb.2019.1367>

This is a good point, and we now include a citation showing that the evolution of coalitionary support is tied to resource monopolizability in primates (Bissonnette et al. 2014).

Lines 218-220: Be more explicit here: the references here refer to within-group inequality in terms of decision-making ability in the context of collective action, right? Once such decision-making hierarchy emerges, it can be more likely to generate inequality in material wealth: <https://royalsocietypublishing.org/doi/full/10.1098/rspb.2014.1349>

Thanks for pointing out this interesting paper, and noting that our claims here were not sufficiently specific. We now specify that we are discussing inequality in influence during collective action, we note that leaders can use this influence to increase inequality in other dimensions of wealth, and we cite the suggested paper.

Lines 254-256: An example here too would be useful- to show how demographic change produces change in mobility. You cite Scheidel: did the Black Death not just lower inequality but also increase mobility?

We elected to use a non-human animal example here, but we have added examples from social perturbation experiments in captive fish, primates, and mice.

Lines 265-267: differentiated social networks may precede the emergence of social classes. Greater social connectivity in networks helps maintain egalitarianism (<https://royalsocietypublishing.org/doi/full/10.1098/rspb.2019.1367>) and greater deviation from panmixia in networks can foster emergence of social classes: <https://academic.oup.com/beheco/article/25/1/58/222376>

These are great examples of how dynamics can give rise to the emergence of social classes, but in this paragraph our goal is to explain the concept of durability of inequality, so we didn't feel like these citations fit well without considerably expanding this section. In light of limitations on length, we elected not to add this discussion of the dynamics that can produce social classes.

Lines 279-281: but wouldn't a faster pace-of-life erode the benefits of competing to be upwardly mobile? One solution is to make behavioral traits associated with such faster pace-of-life facultative, as evident in humans: forms of risk-taking (<https://pubmed.ncbi.nlm.nih.gov/27149981/>) and discounting of the future (e.g. <https://pubmed.ncbi.nlm.nih.gov/28073390/>) covary with SES.

The reviewer might be right about this. Given this sentence was a bit speculative, we simply removed the bit about expecting faster pace of life here.

Referee 3

Overview and general recommendation:

The origin and impacts of inequality in human societies is widely studied, while often only studied in animal societies in terms of dominance or reproductive skew and has rarely been linked to economics, anthropology and psychology. The current paper is an interesting and comprehensive review of inequality in both human and animal societies and aims to integrate the two fields identifying common principles when appropriate.

The paper is structured around four main sections which are clearly presented in Figure 1: a) What is wealth and inequality in animal societies? b) What are the consequences of inequality? Although not defined as such, there is a section on c) Why are societies unequal? Which could also be labelled What are the causes of inequality? to better fit with Figure 1. Finally, there is a section on d) How does inequality change over time?

We have relabelled section 4 as “What are the causes of inequality?” as recommended.

The paper is well researched with an impressive 140 references that are relevant to each section. The paper successfully brings together the knowledge and methods used in human societies and in particular on the inequality of wealth, to draw parallels with animal societies. I am more familiar with the ecology and evolution literature and while most relevant references are cited, I suggest ways below in which they could be better discussed to highlight the work that has been carried out in animal societies. One point which is not explicitly discussed, is that in human societies research has concentrated on inequality in wealth itself, which is access to resources, physical condition and social connections, whereas research in animal societies has concentrated more on the consequences on inequality of wealth, such as inequality in reproductive success and ultimately fitness. I would suggest that this should be more apparent in the two different sections, the strengths of research in human vs animal societies and vice versa as a function of which traits have been measured.

We have extensively edited the introduction and we have highlighted this distinction between inequality in outcome (e.g., health, well-being, fitness) vs inequality in mechanism (i.e., wealth). We are hesitant to draw too stark of a contrast between human and animal literature regarding a focus on wealth inequality vs. outcome inequality because lots of work in humans does also look at inequality in outcomes, either focusing on health (sociology, epidemiology) or on human fitness and evolution (human behavioral ecology, evolutionary anthropology).

I disagree with the authors claim that “very little work in non-humans has explored pathways by which inequality impacts individuals, societies and evolution“ and I have suggested ways in which this could be modified. I have suggested multiple measures of inequality in reproductive success and even

of wealth that have not been mentioned in this paper. Certain areas could be expanded upon including: perception of inequality in animal societies, the altering of social behaviours and conflict resolution.

We agree that this statement was too strong, and have softened this claim by simply calling for more work into the various ways that wealth inequality impacts individuals, societies, and evolution. We thank the reviewer for their constructive and detailed comments.

Major comments:

2. What is wealth and inequality in animal societies? As stated on lines 20-21: Inequality in access to resources, physical condition and sociality (*measures of wealth*) translates into differences in health, longevity and reproductive success and ultimately fitness. Box 1 is mainly concerned with a description of different measures of human inequality in terms of wealth (*income*), but also land ownership, social connection, faculty production and body size. Box 1 also measures inequality in animal societies in terms of reproductive success and body size. However, I would consider reproductive success in animal societies as the outcome of measures of wealth, whereas no fitness measures are described in Box 1 for humans (health, longevity and reproductive success). Reproductive success fits more into **3. What are the consequences of inequality?** Alternatively, Box 1 could be expanded to include measuring inequality in wealth AND measuring inequality in reproductive success (see below).

We agree that disparities in reproduction are best treated as the outcome of wealth inequality rather than wealth inequality itself. We hope this is now clearer in our revamped introduction and throughout the manuscript. We decided that including the application of the Gini coefficient to reproductive success here simply invites confusion, and we have removed it from Box 1.

3. What are the consequences of inequality?

Lines 122-124: I would argue against that “very little work in non-humans has explored pathways by which inequality impacts individuals, societies and evolution” as the vast field of sexual selection in ecology and evolution describes and widely quantifies how “inequality in wealth” i.e. access to resources, physical condition and sociality (*measures of wealth in this paper*) translates into reproductive success. On lines 198-199, the classic paper by Emlen & Oring 1977 is referenced in relation to the behavioural and ecological conditions as drivers of inequality, which is accurate, but a lot of work stemming from this theory discusses the consequences of inequality on mating and social systems.

Classical sexual selection theory predicts that inequality will be higher (the higher the variance in reproductive and mating success) when the more the access to either one of the sexes, or to reproductive opportunities (*material wealth*), is limiting, the stronger will be the competition between individuals of the opposite sex (*embodied wealth*) and the stronger the sexual selection (*inequality*) (Emlen & Oring 1977; Andersson, 1994; Bateman, 1948; Darwin, 1871; Trivers, 1972). The potential for such multiple mating depends on feasibility within an individual’s time budget (little or no parental investment), and whether multiple mates, or resources required for multiple mating (*material wealth*), can be monopolised in time and space (Emlen & Oring 1977). The higher fitness gain from having multiple partners (*inequality*), the sexual selection or Bateman gradient (Bateman, 1947, Arnold & Wade, 1994), is the cause of sexual selection, giving rise to stronger male-male competition, female mate choice, and greater variation in structural and/or behavioral traits in males (*embodied wealth*). Sexual selection determines the resulting mating system (*society*) and explains its evolution.

In terms of measures of inequality of reproductive success, there are many. I have already mentioned the Bateman gradient, but there are also Bateman’s variances (the opportunity of sexual selection I_s and selection, I), the index of resource monopolization (Q) and the Morista index. Furthermore, other measures of sexual selection include selection differentials (s') and selection

gradients (β') that measure the direct selection on phenotypic characters to reveal the target(s) of sexual selection (Lande & Arnold 1983). These coefficients quantify the intensity of sexual selection and have greater predictive value in relation to evolutionary change. A few papers have compared these different measures: Mills *et al.*, Proc Roy Soc Lond B (2007) 274, 143–150, as well as Fairbairn & Wilby (2001) and (Jones et al. 2004, 2005) referenced within.

We have removed the problematic phrasing about “very little work in non-humans have explored...”. With respect to sexual selection as an example of an ecological framework of inequality, we now bring up this point in the revamped introduction. However, we also try hard throughout the manuscript to differentiate between wealth inequality vs. fitness variation, as we are focusing on how the latter is the outcome of the former (as the reviewer mentioned in the previous comment). In the section 4 (“causes of inequality”), we mention various ecological drivers of inequality, many of which were inspired by Emlen & Oring. We also include resource monopolizability and its link to the evolution of behavioral mechanisms such as coalitions. We do not focus on the measures of variance in reproductive success, as this is the down-stream consequence of wealth inequality, and as noted by the reviewer here, this has been very thoroughly covered by the fields of sexual selection and social selection.

Individual perception of inequality: Lines 140-146: There are also mechanisms in place that perceive inequality in animal societies. For example, males falsely signaling their reproductive quality (*embodied wealth*) will either suffer mortality due to the cost of maintaining the signal or injury/death after losing in competition to other males. Zahavi’s handicap principle posits that signals will provide reliable information about the quality of signalers, provided that they are costly to produce (Zahavi 1975, 1977). This sexual selection principal has been proven both theoretically (Enquist 1985; Grafen 1990; Godfray 1991; Maynard-Smith 1991; Johnstone and Grafen 1992) and empirically (e.g., Andersson 1994; Johnstone 1995; Møller 1995; Mappes et al. 1996; Kilpimaa et al. 2004). The expectation of reliability is inherent in both “viability indicator” and Fisherian mechanisms of sexual selection (that are but a continuum of a single process, Kokko et al.

2002) manifest in sons as either superior survivorship and growth or attractiveness, respectively (Greenfield and Rodriguez 2004).

A female’s choice of mate may be based on a signal or other advertisement feature (*embodied wealth*), that is a reliable indicator of a potential mate’s phenotype, and ultimately their heritable fitness (Moore 1994; Welch et al. 1998; Møller and Alatalo 1999; Doty and Welch 2001). The simplest mechanism for the maintenance of signal reliability is physical constraint, such as the carotenoid-based plumage coloration in male house finches, *Carpodacus mexicanus*, which accurately indicates nutritional condition, and thus health or foraging ability (Hill and Montgomerie 1994) and females selecting brighter males acquire higher quality mates (Hill 1991). This is an example of a reliable signal.

In our revisions, we initially added in some discussion of the role of individual perception of other individuals’ wealth as suggested by the reviewer. However, in order to shorten the manuscript to fit the journal length limits, we once again removed this discussion. Our reason for doing so is that, although the reviewer is correct in noting that perception of others wealth is a relevant evolutionary force, it doesn’t relate that strongly to wealth inequality at the group level. To emphasize our novel contributions in response to other reviewer’s comments, we decided that highlighting the multilevel nature of inequality and their effects should take precedence here. We thank the reviewer for their suggestion.

Lines 142-151: An example of altering social behavior in animal societies can be found with subordinate *Polistes* paper wasps show increased aggression to the dominant queen if the subordinate eggs are removed – this represents an increase in reproductive skew with the subordinates receiving a decrease in reproduction and hence they retaliate. Reeve, H.K. & Nonacs, P. (1992) Nature 359, 823-825.

This is a nice suggestion, and we have now included this citation here.

Inequality and group outcomes: Lines 152-170: There are also mechanisms in place in animal societies, to resolve conflicts of inequality. I am not familiar with the human society literature and hence whether there is an equivalent, but it would be interesting to include a section on conflict resolution. In some animal societies, for example in clownfish groups, there is inequality in reproduction with only the two largest individuals reproducing and yet smaller non-reproducing individuals stay within the group and there is no conflict. In this example, group conflict is resolved with the maintenance of size differences between individuals ensuring that smaller individuals, or subordinates, do not become a threat and challenge the reproductive status of the larger or dominant individuals. Here are two relevant publications:

Buston PM (2003) Size and growth modification in clownfish. *Nature* 424:145–146
Wong et al (2016) The four elements of within-group conflict in animal societies: an experimental test using the clown anemonefish, *Amphiprion percula*. *Behav Ecol Sociobiol* 70:1467–1475

To my knowledge, it is not yet known if the subordinates “pay to stay”, a mechanism by which subordinate individuals regulate their own growth so as not to incur eviction and remain queuing within the group. Subordinates “pay to stay” by which subordinate individuals increase cooperative effort, or whether their growth is under social control by the dominant individuals. These are two very different mechanisms and may be compared with cooperation and “manipulation” in humans.

If this is indeed similar to cooperation in humans, evidence of the “pay to stay” mechanism has also been reported in other fish and insects: in cichlids, *Neolamprologus pulcher* (Heg et al., 2004; Bruintjes & Taborsky, 2008), gobies *Paragobiodon xanthosomus*, (Wong et al., 2008) and paper wasps, *Polistes dominula* (Grinsted & Field, 2017). There are dominance hierarchies based on weight in many social mammals (Veiberg et al., 2004) and in other social mammals subordinate females can be aggressively evicted by older dominants (Kappeler & Fichtel, 2012; Pope, 2000). However, no mammalian studies have yet investigated whether individuals modify their growth rates or levels of cooperation to minimize conflict with the dominant.

Thanks for pointing out this literature that indeed fits well within the scope of the paper. Our opinion is that this literature fits best in the “Why are societies unequal?” section, as wealth suppression (either offered voluntarily by subordinates or imposed by dominants) is a mechanism producing wealth inequality among individuals. A parallel example to growth suppression in fish is the interruption of social bond formation in ravens (Massen et al. 2014).

Line 179: **Why are societies unequal?:** To follow the current structure of the paper, this section could be **4. What are the causes of inequality?** Also reference should be made to Figure 1 (bottom left).

Thanks, we have fixed this.

Line 227: **4. How does inequality change over time?** Actually refers to **Dynamics of inequality** and reference should be made to Figure 1 (bottom right).

This now reads “5. How does inequality change over time?” and reference is made to the appropriate panel in Figure 1.

Minor comments:

A personal suggestion I would replace “like” with “such as” throughout: line 70, line 72, line 250, line 334.

We have changed these to “such as.”

Lines 71- 82: The order in Figure 1 and in the summary on Lines 68-71 is Material wealth, Relational wealth and then Embodied wealth. However on lines 71-87, this order is changed, a small point, but

makes reading easier. On lines 71-73 you start with Embodied wealth, Lines 73-78 you shift to Material wealth and finally on lines 78-82 Relational wealth.

We have fixed this so that we discuss these in the same order in all three places.

Lines 71: Embodied wealth in the text should include body size as it does in Figure 1, as well as ornament size, both visual and olfactory and courtship displays

We now reference body size, ornament size, and display quality in the text.

Line 109: I would define WEIRD.

We elected to remove this acronym altogether as it was not necessary.

Line 104: typo, either “an individual” or “individuals”.

Fixed

References

Some references have used a capital letter at the beginning of each word of the title and need to be corrected: references 4, 5, 6, 8, 11, 18, 25, 41, 48, 54, 59, 62, 71, 75, 85, 89, 96, 100, 106, 107, 108, 110, 121, 126, 127, 132 and 135.

Fixed

All Science references have strange section in the title (*80-*). I have seen this before when using Mendeley. See references 6, 15, 31, 40, 50, 82, 94, 132, 134,

Reference 60: is missing journal specifics (number and page numbers) Reference

This paper is from an extra issue with no volume number. We have fixed the omission of page numbers

121: Species name needs to be placed in italics, *Macaca mulatta*

This reference no longer appears in the manuscript

Appendix D

Comments to the Author(s).

I appreciate the effort that the authors took to substantially re-write and re-organize the manuscript. I find this revised manuscript stimulating and can potentially draw the attention of ecologists and evolutionary biologists interested in animal societies to think more broadly and cohesively about inter-individual differences in niche, social relations and phenotypic values under one framework, such as the proposed 'wealth' framework that has been well developed in the research of human societies. I am still struggling a little bit with how operational this proposed wealth framework will be. For example, what would be the appropriate currencies (e.g. energy acquired in optimal foraging theory, inclusive fitness in cooperative breeding) that we could use to measure across different wealth dimensions (material, relational, embodied) and across study systems? That being said, I do agree that there is a need to develop a cohesive framework for studying and understating the interplay between inter-individual differences and animal societies.

Thanks to the reviewer for these constructive and supportive comments. We respond to each in detail below.

Below I provide some suggestions that should help improve the clarity of this review:

1. The ecological and evolutionary unit of wealth inequality is necessarily dependent on the study system and research question. Therefore several different units are mentioned throughout the manuscript, such as population (demographic or genetic), group (as in social animals, which is often based on kinship), society (mostly used in human studies), community (appeared only in one sentence on page 8). While the usage of these related but not identical concepts appears to be appropriate, I suggest adding a glossary section to briefly define each of these units for which the "degree of inequality" can be measured in different contexts. This may help reduce potential confusions across readers from different research fields. For instance, a human society is ecologically closer to an animal population rather than an animal group (which may be closer to an extended family in human?).

This is a great point, but rather than including a long (and potentially contentious) glossary, we instead added a few sentences specifically addressing the multiple levels at which inequality can be measured in the section where we define inequality (Page 7). This definition paragraph now reads: "Wealth inequality describes the spread and skewness of distributions of wealth (Figure 1, center circle) in these different dimensions (Box 1). The scale at which inequality is assessed can be tuned flexibly according to the question and study species. For instance, one can measure inequality among individuals in a society or social group, or among individuals in a population consisting of multiple social groups. When wealth operates at the group level (e.g., group territories, shared food caches), wealth inequality among groups can be assessed at the population level."

2. On a related note, I advise not to include individual as a potential level for measuring inequality. Specifically on page 8: "Inequality at multiple levels (i.e., overall level of inequality of a community, as well as individual's relative position within the community...)", which I think is a bit confusing as inequality is really a measure of the distribution of wealth or fitness across some numbers of individuals (a group, a population, a society; Fig.1, page 7). At individual level, it is a "relative wealth/fitness position" that is the relevant measure here. Because the concept of inequality is logically a positive value (uniform distribution of

wealth/fitness gives the minimum inequality = 0, which can only go up from there), it is difficult to apply a measure of inequality at individual level (do we say that an individual with a lot more wealth or a lot less wealth than other members of the group experiences the same amount of inequality?). I think “relative wealth/fitness position” is a fundamentally different measure from inequality as it carries both a sign (wealthier or poorer) and a magnitude (how much more or how much less).

This is a good point: at the individual level, the relevant measurement is wealth, whereas wealth inequality describes wealth distributions among individuals. To fix this sentence, we now say “Wealth and wealth inequality impact individual health and well-being” at the start of this paragraph, and at the end of the prior paragraph we rephrase the sentence about inequality at multiple levels to read: “the overall level of inequality at the group or society level may have effects beyond an individual’s wealth.”

3. Temporal changes in wealth inequality (social mobility) surely is an interesting aspect of wealth inequality. However, when concerned with evolutionary patterns (e.g. maintenance of wealth seeking behavior) or long-term ecological dynamics (e.g. stability of animal society), should the degree of social mobility be treated as a component of inequality measure (a population with higher social mobility is on average of lower inequality)? A bit clarification would be nice.

We view social mobility as a related measure rather than a component of inequality. To clarify this, we have added to the end of the first paragraph on social mobility: “By integrating over time, social mobility mediates the link between inequality measured at a given time point and the processes or outcomes occurring over individual lifetimes.”

4. Are there also spatial dynamics of social mobility? As the authors pointed out, individuals may choose to leave a group of low temporal social mobility and join another group of higher temporal social mobility, which is well known in human societies. Therefore I would think social mobility involves spatiotemporal dynamics.

There could be a spatial element to social mobility in some cases (such as the example raised by the reviewer), but we focus here on temporal dynamics as the core feature of social mobility, since there is a deeper literature on this topic in both human and non-human societies. Spatial dynamics of may be a topic of interest to address in the future, but we elected not to add a more detailed discussion here since we are not aware of much work on this type of social mobility in non-human animal studies that we can draw from.

5. Three dimensions of wealth (material, relational, embodied) were defined in this manuscript and the concept of multidimensional wealth briefly mentioned (e.g. pages 6. 16). However, it was not clear whether the authors would recommend an approach to quantify inequality in multidimensional wealth space (similar to the Hutchinson’s hypervolume; that is, a population, a society and/or a group may have one overall value of inequality corresponding to a volume in the 3-dimensional wealth space), or to treat these dimensions as separate but linked components (e.g. the amount of inequality in embodied wealth may lead to or covary with that in material wealth)? Note that “multidimensional” does not simply mean “considering several dimensions,” and therefore it is better not to use these two terms

as synonyms. It would be helpful to see some specific scenarios where a truly multidimensional approach will be appropriate and beneficial in studying wealth inequality.

This point is well made. Here, there may be less utility in a hypervolume-like approach. Instead, the primary value of considering these multiple dimensions of wealth is to understand how they relate to each other and have shared causes and consequences. To resolve the terminology issue raised by the reviewer, we have removed the word multidimensional from the paper.

6. I urge the authors to consider including a brief discussion on the possibility that neutral processes such as patchy resource distributions or phenotypic variation due to genetic drift can also lead to wealth inequality (e.g. patchy resources allowing some individuals to have more material wealth than others by chance, genetic drift allowing some individuals to have a larger body size than others by chance). Therefore a null model approach (i.e., comparing the observed amount of wealth inequality against the amount of inequality arose by chance) that is commonly employed in ecological and evolutionary research may also apply here.

We already highlight resource patchiness in the “causes of inequality” section, although we don’t label it a “neutral” process. The null model approach suggested by the reviewer is an interesting idea, but in our opinion the drivers of inequality are not yet well-enough understood to quantify how much inequality would be expected by chance under a particular condition. Therefore we do not feel we could provide a very concrete suggestion for this approach, and elected not to add this discussion to the paper.